



# Introducing a new normalized cryospheric index (NCI) to categorize sub-watersheds on arid environments

Christopher Ulloa[1], Ayon García[1], Anouk Beniest[2]

1LICA-DICTEC/Water and Cryosphere Research Laboratory, University of Atacama, Avenida Copayapu #485, Copiapo, Chile

2 Department of Earth Sciences, Vrije Universiteit Amsterdam, De Boelelaan 1081, 1085 HV, Amsterdam, The Netherlands

Correspondence to: christopher.ulloa@uda.cl

**Abstract.** This study examines sub-watersheds in the arid northern region of Chile (26°41'–28°24'S), situated within the broader Copiapo watershed. The primary water source for this watershed originates from cryospheric reservoirs. The region exhibits pronounced physiographic and climatic diversity, with its northern sector situated within the South American Arid Diagonal (SAAD), where cryospheric features exhibit greater spatial isolation. The aim of this study is to quantify the water volume contributed by distinct cryoforms to regional watersheds. This study employs a classification methodology to categorize cryospheric reservoirs within sub-watersheds, integrating an inventory of cryoforms, historical snow cover data derived from satellite imagery, and published ice content and depth measurements. The Normalized Cryospheric Index (NCI) is calculated under varying hydrological conditions to assess and compare potential water volumes across sub-watersheds. The analysis reveals significant spatial variability in cryospheric reserves and their strategic hydrological significance. Under average and low-precipitation conditions, the southern sub-watersheds of the Copiapo river Basin exhibit the greatest water storage potential. The Montosa river (NCI = 0.82), Manflas river (NCI = 0.62), *Estero Come Caballo* (NCI = 0.57), and Del Potro river (NCI = 0.51) sub-watersheds have been identified as strategic priority areas within the region for sustaining surface runoff and safeguarding water availability. During high-snowfall periods, northern sub-watersheds in the Copiapo river Basin, such as *Estero Come Caballos*, exhibit elevated NCI values despite their limited cryospheric reserves. In contrast, the Montosa, Manflas, and Pulido sub-watersheds contain the most extensive cryospheric reserves and rank among the top four sub-watersheds with the highest NCI scores.

## 1 Introduction

Anthropogenic climate change impacts water availability, particularly in regions dependent on high-altitude water sources within mountain systems (Beniston & Stoffel, 2014, Beniston et al., 2018). Shifts in the cryosphere that will occur within the next century include a change from solid to liquid precipitation, retreating cryoforms and snow lines to higher altitudes, and shorter snow seasons, predicting that water resources in mountain areas will become increasingly scarce. In the arid zones of the Chilean Atacama Desert glaciers are in constant retreat (Masiokas, et al., 2020) and centennial reconstructions of rainfall patterns on the scale of hundreds of years, show a decline in rainfall in the Central Zone and the "Norte Chico" natural regions





of Chile (Le Quesne et al., 2006; Minvielle and Garreaud, 2011; Morales et al., 2012; Bozkurt et al., 2017). In addition, the
occurrence of mega droughts has intensified since 2010 (Garreaud et al., 2019).
The Copiapo watershed (Fig. 1) is located in the southern limit of the Atacama Desert, near to the South American Arid
Diagonal or SAAD (Zech et al., 2008) in the driest desert of the world (Clarke, 2006). The annual precipitation in Copiapo
from 1795 to the present shows a mean annual value of 22.5 mm (Izquierdo et al., 2024). Among the arid basins in northern
Chile, the Copiapo watershed is regarded as one of the most vulnerable (Suárez et al., 2014). Because of the limited availability
of freshwater and the overexploitation of superficial water and groundwater, the deepening of the water table is accelerating
and cryospheric reserves and resources are reducing (Casassa et al., 2007; Vuille et al., 2008). Despite this very limited liquid
precipitation in the lower areas, e.g. around the city of Copiapo, there is a flow of water in the rivers of the Copiapo watershed
year round. This water comes from the Andean cryosphere that is known to host and store large amounts of fresh water (Bolch
and Marchenko, 2006; Bórquez et al., 2006; Azocar and Brenning, 2010; Schaffer et al., 2019).
Because of the high altitude of the Andean cryosphere, precipitation is captured from air masses, predominantly in the form
of snow. Generally, snow precipitation occurs during the winter (June-September), although the northern parts also receive
snow during summer due to the Tropical Monsoon (i.e. Bolivian winter), which takes place from December to February
(Valdivieso et al., 2022). This snowfall can transform into cryospheric reserves that are expressed in cryoforms such as
uncovered glaciers (debris-free glaciers), debris-covered glaciers, and ice-rich permafrost cryoforms. The melting of these
cryospheric systems generates a sustained discharge of water, even during snow-free dry seasons.Given the limited amount of
rainwater in the Andes, the main source of water entering the Copiapo river comes thus from melting cryospheric reserves at
higher altitudes during spring and summer and these are responsible for maintaining runoff in dry years, while in wet years it
is the snow or cryospheric resources that produces most of the runoff (Ohlanders et al., 2013). Sublimation in the melting
process can reach up to a 70% loss of the snowpack (Jara et al., 2021), but still the snow that falls and melts dominates the
streamflow.
In years with limited snowfall, landforms composed of ice, rocks and debris function as cryospheric water reserves (Meier,
1969). These cryoforms are the main source of water in the Copiapo watershed. Ice-rich permafrost develops extensively in
the upper part of the Copiapo watershed in cryoforms such as protalus lobes, gelifluction taluses (i.e. slopes with ice-rich
permafrost) and rock glaciers (Garcia et al., 2017).
The source of water in the watersheds of the Atacama Desert originate thus predominantly from the volumes of ice that are
kept in cryospheric reserves, but the potential volumes of water that are present in these cryospheric reserves are highly variable
and depend on both the type of cryoform and the spatial distribution. We hypothesize that water is unevenly distributed across
the Atacama Desert, depending on the types of cryoforms and their specific locations, which implies that the potential water
supply may vary accordingly. Consequently, once the ice bodies begin to melt, significant variability in water runoff is
expected throughout the region.
General methodologies to classify and prioritize watersheds are based on morphometric analysis (Rahaman et al., 2015),
watershed hydrology (Wolfe, et al., 2019) and ecological and water quality factors. There are also numerous studies covering



watershed health in which protection frameworks have been developed based on ecological and hydrological parameters
providing approaches on watershed prioritization (Jaiswal et al., 2015; Ahn & Kim, 2017; Sriyana, 2019; Basuki et al., 2022).
Few studies incorporate all cryospheric components into watershed management or conservation strategies. Peng et al. (2022)
developed an integrated index to evaluate cryospheric changes across the Northern Hemisphere, emphasizing the influence of
cryospheric dynamics on regional hydrology. Nonetheless, there is currently a lack of research addressing cryospheric reserves
and resources within arid watershed contexts.
The main objective of this study is therefore to develop a methodology that is able to identify and integrate the cryospheric
components into a quantitative tool that can monitor the potential volumes of water of each sub-watershed. A sub-watershed
is a watershed unit with a lower Strahler stream order than the main watershed. In our research area, the Copiapo watershed is
the main watershed with rivers that have a maximum Strahler stream order of nine. Within the Copiapo watershed, the
cryospheric sub-watersheds typically contain rivers that originate at the headwaters with Strahler stream orders of one to three
and progress downstream to lower areas, where they culminate in sub-watersheds featuring rivers with a Strahler stream order
of up to nine. By distinguishing among sub-watersheds and their associated cryoforms, we are able to perform both quantitative
and qualitative assessments of the current potential water volumes within cryospheric watersheds. Our case study focuses on
the Copiapo watershed, which encompasses 12 sub-watersheds (Fig. 1, Table 1). These sub-watersheds are situated within the
mountainous catchment area of the main basin and were selected based on the presence of both cryospheric reserves and
resources. Since there is currently no regulatory framework in Chile on watershed management and conservation, that includes
potential water volumes from cryospheric reserves, this study includes a technical workflow on which policies for the
regulation of cryospheric watersheds can be based. The hydrological role of these cryospheric watersheds will prove an
important factor for sustainable water resource management for the Copiapo watershed, especially when the freshwater
reserves currently captured in high altitude cryoforms become available for the Chilean population in the very near future due
continuously increasing global temperatures (Flores et al., 2018).
In this study, we propose the introduction of a "Normalized Cryospheric Index" (NCI) as a novel framework for "cryospheric
watershed classification." The NCI will enable the quantification of potential water volumes stored in cryospheric reserves
and resources, specifically, the distribution and persistence of snow occurrences, within each sub-watershed of the Copiapo
watershed. This cryospheric watershed classification will complement the existing classifications of pluvial and snow-
dominated watersheds (Whitaker et al., 2008; Sanmiguel et al., 2017), which are inadequate for watersheds primarily fed by
the melting of cryospheric reserves.



**Figure 1. Location map of the Copiapo watershed and the different sub-watersheds (Zone 19S). The codes in each sub-watershed represent the official numbering that the General Water Directorate assigns to each sub-watershed. Source map provider: National Geographic, Esri, Garmin, HERE, UNEP-WCMC, USGS, NASA, ESA, METI, NRCAN, GEBCO, NOAA, increment P Corp.**





104

**Table 1. List of sub-watersheds with cryospheric components in the study area.**

| Sub-watershed Code | Sub-watershed name | Sub-watershed area km$^2$ |
|---|---|---|
| 3400 | Quebrada Monardes | 803,6 |
| 3401 | Figueroa river | 924,2 |
| 3402 | Estero Come Caballos | 880,6 |
| 3403 | Cachitos river | 734,1 |
| 3410 | Vizcachas de Pulido river | 611,9 |
| 3411 | Ramadillas river | 366,8 |
| 3412 | Del Potro river | 470,6 |
| 3413 | Montosa river | 413,2 |
| 3420 | Manflas river | 728,1 |
| 3440 | Quebrada San Andres | 1476,3 |
| 3441 | Quebrada Paipote | 1493,8 |
| 3443 | Quebrada Martinez | 1513,1 |

## 2 Regional setting

The cryosphere of the study area is composed of both cryospheric reserves and cryospheric resources. In this study, we differentiate between cryospheric reserves and resources, because both reserves and resources contribute differently to the potential volume of water in cryoforms. The cryospheric reserves of the Atacama Desert host year-round stable cryoforms (Fig. 2), such as debris-free glaciers, debris-covered glaciers, rock glaciers, gelifluction taluses, and protalus lobes, as documented by Richmond (1952). Together, these five classes comprise the cryospheric reserves found in the desert regions of northern Chile and Argentina, whose classification and characteristics were clearly defined by García et al. (2017). The cryospheric resources correspond to the snow fraction that produces freshwater by melting in the same year in which it precipitates.

The glacial and periglacial inventory of the Copiapo watershed indicates a transition from areas that are dominated by glacial environments in the south, to areas that are dominated by periglacial environments in the north (García et al., 2017). In the Copiapo watershed, the occurrence of snow begins above 3,500 m of altitude. In these extreme environments, studies suggest that the water budget is significantly influenced by snow sublimation. From 2001 to 2016, approximately 70% of the snow balance was lost to sublimation, effectively reducing the available water volume (Jara et al., 2021). In semi-arid regions of





120    Chile and Argentina, snow accumulation and subsequent melt processes contribute substantially to surface runoff in numerous

121    Andean river watersheds (Masiokas et al., 2006; Favier et al., 2009).

122

**Figure 2. Main cryoforms. Field photographs from the Copiapo watershed on the left, with a schematic interpretation on the right of. a) Debris-free glacier and debris-glaciers, b) Rock glaciers of the Del Potro river sub-watershed. c) Gelifluction taluses of Montosa river sub-watershed, d) Protalus lobes of the La Laguna sub-watershed in Pulido watershed.**





## 2.2 Hydrological background of the cryosphere: cryospheric ice volume estimations

The potential volume of water that is captured in cryoforms of a cryospheric watershed refers to its capacity to produce water by melting its cryospheric reserves and resources, i.e. debris-free glaciers, ice-rich permafrost and seasonal snow. Assessing water productivity in cryospheric watersheds presents a significant challenge, with limited research attempting to quantify the water yield from distinct cryospheric classes or to evaluate the capacity of glacial and periglacial landforms to generate and retain these water volumes (Ayala et al., 2020). Existing studies predominantly concentrate on exposed glaciers and rock glaciers, which will be briefly examined in this review.

One of the earliest works on Andean permafrost was done by Corte (1978). In this study, Corte (1978) compared the water contributions of debris-free glaciers and rock glaciers. Corte (1978) discovered that 56% of the total annual runoff of the Cuevas river in the Mendoza province of the Argentinean Andes, located between the towns of Portillo and Uspallata, originated from rock glaciers, while water originating from debris-free glaciers contributed to only 44% of the total discharge. A comparable investigation was conducted for the Maipo River, revealing that in the absence of snowfall, exposed glaciers contribute as much as 67% of the total discharge (Peña and Nazarala, 1987). Similarly, for the San Juan River, Milana (1998) proposed that meltwater from both exposed and debris-covered glaciers accounts for up to 70% of the river's total runoff. Croce and Milana (2002) concluded that rock glaciers on the Argentinean side contribute to the surface runoff with an estimated stored water volume for the "El Paso Rock Glacier" (30°130S, 69°480W) of $6.3 \times 10^6$ m$^3$, releasing water in times of drought. Other studies were conducted to determine the volume of ice hosted in rock glaciers (Barsch, 1996; Burger et al. 1999), concluding that in the Alps about 50% of the volume of rock glaciers consists of ice.

**Table 2. Water productivity of different watersheds of the Copiapo watershed. Modified from Schaffer et al (2019).**

| Location | Latitude | Hydrological year | Contribution to streamflow (%) | Reference |
|---|---|---|---|---|
| Huasco watershed | ~ 29°S | 2003/04 and 2007/08 | 3–23 | Gascoin et al (2011) |
| Elqui watershed | ~ 30°S | 2003 | 4–9 | Favier et al (2009) |
| Elqui watershed | ~ 30°S | 2011 | 13 | Pourrier et al (2014) |
| Juncal river Watershed | 32–36°S | 2005/06 and 2008/09 | 10 and 31 | Ragettli and Pellicciotti (2012) |
| Juncal river Watershed | 32–36°S | 2005/06 and 2008/09 | 16 and 44 | Rodriguez et al (2016) |
| Juncal river Watershed | 32–36°S | 2011–2012 | 2-~ 50 | Rodriguez et al (2016) |
| Yeso river Watershed | ~ 33.5°S | 2014/15 | ~ 42 and ~ 67 | Ayala et al (2016) |
| Central Chilean Andes | - | 1968/69 | 67 | Peña and Nazarala (1987) |



For the arid Chilean Andes, Azocar and Brenning (2010) investigated the hydrological significance by comparing debris-free
glaciers and rock glaciers in a central segment of the Chilean Andes (27°-33°S). They modeled that an equivalent of 2.37 km$^3$
of water was present in the rock glaciers. They concluded that, at least in the most arid section of the Andes, the rock glaciers
are a more important frozen water reservoir than the debris-free glaciers. Azocar and Brenning (2010) lay the foundation for
this study by stating that the potential volumes of water in periglacial cryoforms play a fundamental role in the water
productivity of arid catchments. They demonstrated the potential hydrological significance of ice-rich permafrost for water
reservoir estimations. Schaffer et al (2019) have summarized the main water contributions of glacier and rock glaciers to the
overall runoff of different Chilean semi-arid watersheds (Table 2). They demonstrate that glaciers within arid watersheds
contribute 3% to 50% of water to the streamflow in Chile. Glaciers within the Huasco River Basin, the closest neighboring
watershed to Copiapo, account for 23% of the total streamflow in the region.
Trombotto et al. (2020) conducted the inaugural cryo-hydro-chemical assessment of the Stepanek rock glacier in Argentina's
Vallecitos river watershed. Their findings demonstrated that subsurface water and groundwater flow through the cryoform's
active layer. The study revealed that water infiltrates the internal structure of glaciogenic rock glaciers rather than
circumventing it, exhibiting a discharge rate of 71 L/s. This hydrological behavior was linked to possible contributions from
meltwater stored within these glacial formations. Halla et al. (2021) concluded that the Dos Lenguas glacier has an ice content
of 1.71 ($\pm$ 42%) - 2 ($\pm$ 44%) × 10^9 kg with an interannual water exchanges of -36 mm yr-1 (-8.92 × 106 kg) and 28 mm yr-1
(6, 64 x 106 kg). In this case, this represents an important long-term freshwater reservoir, with a spring discharge of 2% to 4%,
0.36-0.43 × 109 kg from the rock glaciers, at the end of the melting period.
Towards the central zone of Chile in the semi-arid Central Andes the continuity of the glacial and periglacial environment can
be observed more evident, for example there is more morphological evidence of the presence of detritus-covered glaciers in
the same cirque as a rock glacier, Monnier and Kinnard, (2017) have established the parameters that regulate the evolution
from glacial to periglacial cryoforms, or from detritus-covered glaciers to rock glaciers. This scenario is little evident in the
Copiapó river basin from the geomorphological approach there is no evidence to infer this continuity, associating the rock
glaciers of this basin mainly to the periglacial environment.
The ice content of gelifluction taluses and protalus lobes remains understudied and is not yet well understood. Hilbich et al
(2022) conducted geophysical studies and borehole logging on rock glaciers, "protalus rampart", gelifluction slopes, colluvial
slopes and talus slopes in Norway, classifying different slope cryoforms. Based on those analyses they confirmed the existence
of ice-rich layers in several cryoforms that were not precisely rock glaciers. They also pointed out that even thin, ice-rich layers
on permafrost slopes can add similar volumes of ice to watersheds where there are individual rock glaciers. With this study,
they emphasized the importance of ice-rich permafrost in the hydrological cycle. While Hilbich et al. (2022) acknowledge that
subsurface ice cannot be identified through visual inspection alone, they emphasize that remote sensing offers valuable first-
order approximations. According to the same study, rock glaciers in the Andes contain between 35% and 55% ice.





**3 Methodology**

**3.1 Study Area**

The Copiapo watershed is composed of 27 sub-watershed, of which 12 are cryospheric watersheds (Table 1), meaning that they store freshwater in cryospheric resources and reserves.

The study area was identified based on the cryospheric components, retrieved from the inventory of glaciers and periglacial environment of the Atacama Desert (García et al., 2017). After mapping, we compared the cryoform polygons to the global permafrost model (Gruber, 2012) and 97% of the mapped cryoforms in this study fall within the high probability range of the permafrost model. The average altitude of these cryospheric watersheds is 3,617 m above sea level. The identification of the 12 cryospheric watersheds of Copiapo watershed was based on data from the glacial and periglacial inventory (Garcia et al., 2017) and snow resources determined using satellite imagery from the MODIS sensor (Rittger et al., 2013).

**3.2 Survey of cryospheric water reserves**

To quantify the ice volume of cryospheric reserves, we used the inventory of glaciers and periglacial environment of the Copiapo river watershed (García et al., 2017) and the watershed data obtained from the world consensus for the determination of ice thicknesses on debris-free glaciers (Farinotti et al., 2019). For the García et al., 2017 inventory, remote sensing techniques, geographic information system (GIS) software and geomorphological mapping techniques were used to remotely detect the presence of active cryoforms along the entire headwaters of the Atacama region.

The area (A, in km²) and perimeter (P, in km) were measured for each landform. The main error-prone parts were the resolution of the data and contour demarcation accuracy. The same approximation formula (Eq. 1) from Paul et al. (2013) was applied uniformly to all cryoforms. This method assumes a one pixel wide "ring" along the perimeter, weighted by $1/\sqrt{2}$, which gives an estimate of mapping uncertainty (Paul et al., 2015).

**3.3 Survey of cryospheric resources: MODIS and snow monitoring**

The MOD10A1 product with fractional snow cover from NASA was used for the cryospheric resources survey. The daily images from the year 2000 to 2022 were processed to evaluate the snow permanence at sub-watershed level. The fractional product of MOD10A1 is an index that classifies each pixel of the satellite image with a snow fraction (Rittger et al., 2013). This product provides an optimized spectral mixing algorithm indicating the fractional amount of snow, i.e. the percentage of the pixel (500 x 500m) that is covered by snow. Note that this model has an accuracy of about 90% under ideal conditions (Hall et al., 2006; Jiang et al., 2024), because cloud cover is sometimes interpreted as snow. Nevertheless, given that the primary aim of this study was to establish a foundational historical snow cover extent capable of delivering a rapid evaluation of snowfall patterns for the preliminary categorization of cryospheric resources, we conclude that this model is appropriately suited for integration into our analytical framework. In this way, the snow cover data in km$^2$ at the daily level were integrated





for the entire analyzed period. The resources areas were calculated using the average snow permanence from zero to 99% for
the entire period and in each sub-watershed.
**3.4 Field work for ground truthing**
Since the mapping, workflow is mainly based on remote sensing, satellite imagery and existing databases, we have conducted
fieldwork to get observations that allow ground truthing of the mapped cryoforms. The analyzed units comprised the sub-
watersheds of the Montosa river (03413) and Del Potro river (03412), representing approximately 10% of the total watershed
area. During field surveys, photographic documentation was conducted upon the identification of specific cryoforms to support
the validation of cryospheric reserve mapping. Given the low surface detectability of these cryoforms, ground-truthing of the
mapped cryospheric reserves was feasible exclusively within the Montosa (03413) and Del Potro (03412) sub-watersheds.
**3.4.1 Geophysical survey on gelifluction slopes**

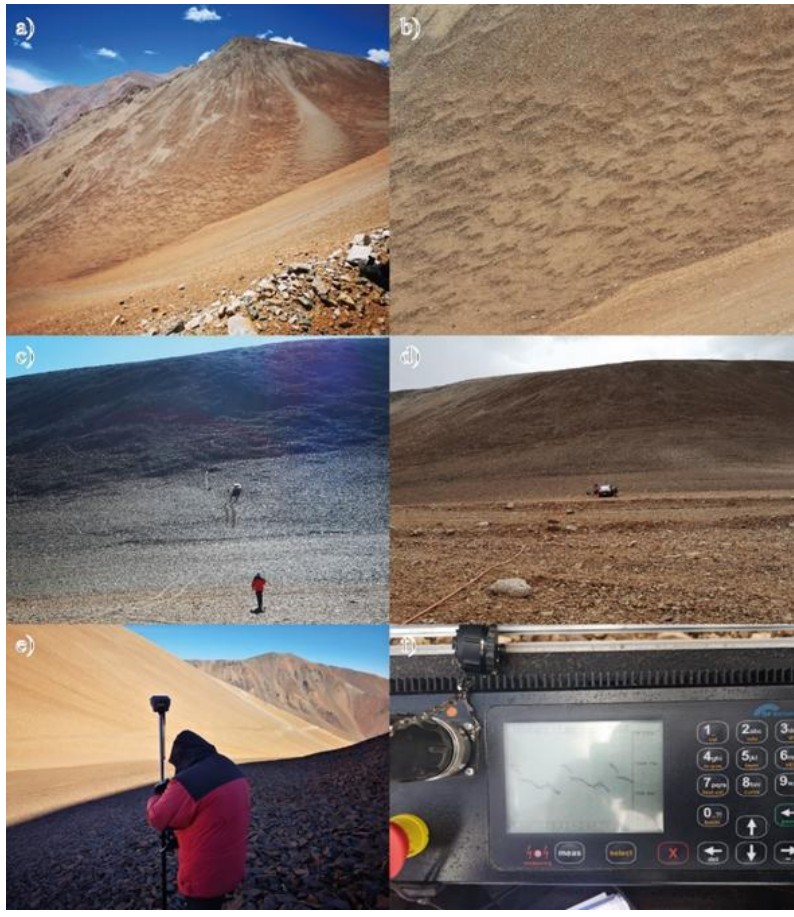


**Figure 3. Photography of the fieldwork. a) And b) The Lobulated morphology of studied gelifluction taluses, which are not covered**
**in snow c) and d) the geoelectric line cable, e) and f) the used equipment, differential GPS and resistivity meter ARES II measuring**
**a resistivity point diagram distribution.**



The ERT is a multi-electrode resistivity method that produces 2-D vertical profiles of the subsurface. The field setup of the
equipment (Table 3) that forms the array includes 36 electrodes for Wenner, Schlumberger and Dipolo-Dipolo arrays (Fig. 2).
The datum points are distributed in pseudo depths to measure the distribution of resistivity of different layers in the subsurface.
**Table 3 Used geophysical equipment for fieldwork of this study**

| Equipment | Amount |
|---|---|
| Resistivimeter: ARES II | 1 |
| 30 electrode box | 5 |
| 10 chanels multicable II (60 m) | 10 |
| Car Bateries | 2 |
| Diferential GPS | 1 |

The ERT array requires a square configuration composed of multicables arranged at 90-degree angles. The square was build
using a "South Galaxy" differential GPS (Figure 1, e) to limit the installation error in the 600 m of profile measurement,
because the line need to be straight, in which the begin and end coordinate were set. Each line was build using 30 m (horizontal
distance) spacing marks to ensure straightness of the sides of the square. The final position of the electrodes was determined
using a measuring tape in combination with the GPS marks installing electrodes every 5 meters. The electrodes were then
connected to the ARES II controller by the multi-electrode intelligent cable of the GF Instruments brand (Fig. 3, f).
Before every measurement, all electrodes were tested to detect electrodes with high standard deviation in resistivity. If the
deviation exceeded acceptable limits, the electrodes were repositioned until the deviation fell within an acceptable range to
ensure measurement accuracy.

**3.5 Calculation of the ice volume and uncertainty**

To quantify the total ice volume stored in the glacial and periglacial cryoforms of the Copiapo watershed, we applied cryoform-
specific approaches to estimate thickness and ice volume, considering both physically – based models and empirical
formulations (Table 5). Each method was selected based on the cryoforms's dynamics, morphology, and availability of relevant
input data. The method applied and associated uncertainties are detailed below.

**3.5.1 Debris-free glaciers**

For debris-free glaciers, we used the physically-based model proposed by Farinotti et al., 2017, which estimates local ice
thickness by inverting Glen´s flow law under the assumption of mass conservation. The ice thickness h was computed as:
$$h = \left[\frac{(n+2)\cdot q}{2A\cdot f\cdot \rho\cdot g\cdot \sin(\alpha)}\right]^{\frac{1}{n+2}} \tag{2}$$



Where $q$ is the specific ice volume flux, $A$ is the rate factor for Glen´s flow law, $n = 3$ is the exponent, $f = 0.8$ is the valley
shape factor, and $\alpha$ is the slope derived from digital elevation model (DEM). Ice volumes were calculated by spatially
integrating ice thickness across the glacier outlines extracted from the Randolph Glacier Inventory v6.0 (RGI Consortium,
2017) for the Copiapó watershed. Following Farinotti et al. (2017), we adopted a relative uncertainty of ±26% on the total ice
volume estimates.

### 3.5.2 Debris-covered glaciers

For debris-covered glaciers, where surface dynamics are significantly decoupled from subsurface ice due to the insulating
effect of debris, we used an empirical area-volume scaling relationship derived by Gärtner-Roer et al. (2014):

$$V = c \cdot A^y \tag{3}$$

With $c = 0.0365$ and $y = 1.375$. Ice content was assumed to range between 45% and 99% of the total volume depending on
the debris thickness and inferred thermal regime. A conservative uncertainty range of ±35% was applied, consistent with the
propagation of scaling law variability and digitalization error.

### 3.5.3 Rock glaciers

For rock glaciers, we applied a physically-based formulation following the shear stress approach of Cicoira et al. (2020). The
basal shear stress $\tau_i$ was estimated using:

$$\tau_i = \tau_{ref} \cdot \left(\frac{\Delta h_i}{L_i}\right)^\gamma \cdot \left(\frac{A_i}{A_{ref}}\right)^\delta \tag{4}$$

And the mean ice thickness $H_i$ was then calculated as:

$$H_i = \frac{\tau_i}{\rho \cdot g \cdot \sin(\alpha_i)} \tag{5}$$

Where $\Delta h_i$ is the elevation difference between the head and rock glacier front toe, $L_i$ is the length of the main axis of each
cryoform (determined manually) (Fig. 7), $A_i$ is the planimetric area, and $\alpha_i$ is the effective slope. We used $\tau_{ref} = 100$ kPa,
$A_{ref} = 0.1$ km$^2$, $\gamma = 1.0$, and $\delta = 0.25$, assuming steady-state deformation. Ice content was assumed to range from 40% to 60%,
based on cryospheric background, and uncertainty was estimated via Monte Carlo simulation.
To account for the uncertainty associated with the ice content of rock glaciers, we implemented a multi-source Monte Carlo
simulation framework. In each iteration, the ice volume was computed using a physically based thickness model, while



randomly sampling the ice content (e.g., 40-60%), reference shear stresses and scaling factors of applied empirical relationships
Cicoira et al., (2020) from a uniform distribution within the plausible range reported in the literature (. The volume of ice $V_{ij}$
for the $i^{\text{Th}}$ rock glacier in the $j^{\text{Th}}$ iteration was calculated as:

$$V_{ij} = h_i \cdot A_i \cdot IC_j \tag{6}$$

Where $h_i$ is the estimated ice thickness (m), $A_i$ is the mapped area of the rock glacier in m² (García et al., 2017), and $IC_j$ is the
randomly selected ice content value for iteration$j$, sampled from a uniform distribution between 0.4 and 0.6.
The final ice volume estimate for each glacier was computed as the mean of all simulated values ($\bar{V}_i$), and the associated
uncertainty was expressed as the standard deviation ( $\sigma_{V_i}$) across all iteration:

$$\bar{V}_i = \frac{1}{N} \sum_{j=1}^{N} V_{ij}, \tag{7}$$

$$\sigma_{V_i} = \sqrt{\left(\frac{1}{N-1} \sum_{j=1}^{N} \left(V_{ij} - \bar{V}_i\right)^2\right)} \tag{8}$$

**Table 4. Parameter ranges used in the Monte Carlo simulation for ice volume estimation in rock glaciers.**

| Parameter | Symbol | Distribution | Range / Value | Units | Reference / Justification |
|---|---|---|---|---|---|
| Reference shear stress | τ\tau | Uniform | 80 – 120 | kPa | Cicoira et al. (2020); reflects site variability |
| Elevation difference | Δhi\Delta h_i | Fixed (measured) | Individual per landform | m | Derived from DEM |
| Flowline length | LiL_i | Fixed (measured) | Individual per landform | m | Digitized from high-res imagery |
| Planimetric area | AiA_i | Fixed (measured) | Individual per landform | km² | Derived from polygon outlines |
| Area scaling exponent | δ\delta | Uniform | 0.20 – 0.30 | dimensionless | Cicoira et al. (2020); conservative range |
| Morphological scaling exp. | γ\gamma | Uniform | 0.80 – 1.20 | dimensionless | Based on model sensitivity range |
| Ice density | ρ\rho | Fixed | 900 | kg m⁻³ | Common assumption for permafrost ice |
| Surface slope | αi\alpha_i | Fixed (measured) | Individual per landform | radians | Derived from DEM and axis |
| Gravitational acceleration | g | Fixed | 9.81 | m s⁻² | Physical constant |
| Ice content | ICj | Uniform | 0.40 – 0.60 | Fraction (0–1) | Schrott (1996); Cicoira et al. (2020) |





| Parameter | Symbol | Distribution | Range / Value | Units | Reference / Justification |
|-----------|--------|--------------|---------------|-------|---------------------------|
| Mapping resolution | RR | Fixed | 0.005 | km | 5 m satellite imagery |
| Perimeter (for ΔA) | PP | Fixed (measured) | Individual per landform | km | Derived from polygon outlines |

This formulation allows us to incorporate not only internal variability (ice content), but also parameter uncertainty within the shear stress model and digitalization uncertainties (e.g. central axis length, slope angle) (Table 4). Additionally, mapping uncertainty in the glacier outlinea was included following the approach of Paul et al. (2015), using:

$$\Delta A = \frac{P}{\sqrt{2} \cdot R} \tag{9}$$

Where $P$ is the landform perimeter and $R$ is the spatial resolution of the delineation (0.005 km for 5-m imagery). This estimate represents a one-pixel-wide uncertainty band around each cryoform boundary.

### 3.5.4 Protalus lobes and gelifluction slopes

For cryoforms with limited thickness and low deformation such as protalus lobes and gelifluction taluses, we used fixed thickness ranges based on literature compilations (Schrott, 1996; Hilbich et al., 2022). Mean ice thickness values were assumed as follow:

- Protalues lobes: mean thickness = 10-20 m, with an ice content of 25-49%.
- Gelifluction slopes: mean thickness = 5-10 m, also with 25-49% ice content.

These ranges reflect published geophysical studies across periglacial environment in the European Alps and Andes. The associated uncertainty was conservatively estimated at ±40%, encompassing morphological variability and interpretation bias in delineating these cryoforms.





**Table 5. Empirical equations used for the calculation of ice volumes of cryospheric reserves.**

| Cryoform | Thickness and volume equation | Ice content | References | Observations |
|---|---|---|---|---|
| Debris-free Glacier | $h = \sqrt[n+2]{\dfrac{q}{2A} \cdot \dfrac{n+2}{(f\rho g \sin\alpha)^n}}$ | 100 | Farinotti et al., 2017 | h = glacier ice thickness, q = specific ice volume flux, A = flow rate factor, n = Glen's flow law exponent, α = surface slope, $f$ = factor for valley shape |
| Debris-covered Glacier | $V = c \times A^y$ | 45-99 | Gärtner-Roer., et al., 2014 | V = Volume, c=0.0365, y=1.375 |
| Rock Glacier | $Hi = \dfrac{\tau i}{\rho g \sin(\alpha i)}$ | 40-60 | Cicoira et al., 2020 | Hi = rock glacier thickness, τi = basal shear stress, ρ = 900 kg/m³ is the assumed density of ice, $g$ = 9.81 m/s² is gravitational acceleration, $\alpha i$ = is the mean slope angle of the landform, expressed in radians. |
| Protalus lobes | max: 5 m, min: 1 m; max: 30 m, min: 20 m | 25-49 | Hilbich et al., 2022; Schrott 1996 | It corresponds to the mean of the data found in this cryoform. |
| Gelifluction Taluses | max: 5 m, min: 1 m; max: 30 m, min: 20 m | 25-49 | Hilbich et al., 2022; Schrott 1996 | It corresponds to the mean of the data found in this cryoform. |

### 3.6 Reserve and resource index and data integration

To develop a classification system that categorizes the capacity to store ice as a strategic water reserve in desert areas of sub-watersheds, we followed a workflow that consists of four phases (Fig. 4). We developed the classification system using cryospheric reserves and resources, integrating both reserves (expressed in gigatons (Gt), equivalent to 1,000 million metric tons of ice) and resources (quantified in km² of snow-cover extent) into the methodological framework. The strategic Normalized Cryospheric Index (NCI) captures the relative significance of each sub-watershed, considering both its ability to store water as ice (cryospheric reserves) and its potential to regenerate water resources through snowfall (cryospheric resources).The NCI needs to be adapted to each specific case. For the Copiapo watershed, the active cryoforms present in cryospheric reserves until 2024 have been used. Those cryospheres include debris-free glaciers, debris-covered glaciers, rock glaciers gelifluction talus and protalus lobes. Depending on the types of cryoforms available in a specific region, the number of cryoforms included may vary.

The methodological framework for computing the NCI index integrates Monte Carlo simulation with a robustness assessment of a composite indicator that incorporates ice volume (V) and snow permanence (S), adhering to the uncertainty propagation principles in simulation-based methodologies outlined by Rubinstein and Kroese (2016).. First, each watershed has a mean value (e.g., $\_V_i$ y $\_S_i$) and an estimate of its dispersion ($\Delta V_i$ and $\Delta S_i$), usually understood as standard deviation. On the basis



that both V and S follow normal distributions $N(\_(V\_i), \sigma 2v)$ y $N(\_(S\_i), \sigma 2s)$, N replicates of each variable are generated
for each watershed, thus obtaining "clouds of values" representing the uncertainty of each datum. Subsequently, a
normalization is performed (using min-max) in order to avoid that one variable dominates the other only by differences in
magnitude.
Once the simulated values are generated, a weighted index is defined whose objective is to combine both variables (Eq. 8):

$$NCI_{i,k}(\omega) = \omega - V_{i,k} + (1 - \omega) - S_{i,k} \tag{9}$$

Where $\omega \in [0,1]$

Where $\omega$ is a parameter that determines the relative relevance of ice volume versus snow permanence. For each sample k
(among the N simulations), the values of that index in all the watersheds are obtained and ordered from highest to lowest,
giving rise to a specific ranking of each run. In this way, each watershed "i" gets a "position" (1 for the watershed with the
highest index, 2 for the next one and so on).
The optimization function (Eq. 9) relies on the stability of the ranking under uncertainty, in accordance with the sensitivity
analysis framework outlined by Saltelli et al. (2008). Specifically, for each watershed, the standard deviation of its position
over the N runs is measured, and this measure of dispersion is summed (or averaged) for all watersheds. A smaller value of
this sum indicates greater stability or robustness in the ranking. Formally, if $\sigma_i$ denotes the standard deviation of the position
of watershed under a weight $\omega$, the objective function results:

$$f(\omega) = \sum (i = 1 \ to \ n) \ [\sigma_i \cdot (\omega)] \tag{10}$$

Where the idea is to minimize the value of this function. To determine the "optimal" $\omega$, a sweep over [0.1] is performed. The
optimal result corresponds to the $\omega$ value that maximizes the stability of the watershed ranking system, ensuring minimal
sensitivity to errors or noise in the V and S datasets.
Finally, plots are generated to analyze the resulting average ranking and the probability that certain watersheds occupy the first
place in various simulations. In this way, a picture is obtained of how uncertainty in ice volume and snow permanence affects
the prioritization of watersheds.





Figure 4. Flow chart of the methodology used to calculate the NCI.



## 4 Results

### 4.1 Glacier and Periglacial Environment Inventory Results

Mapping the cryospheric reserves inventoried 1862 cryoforms in the Copiapo watershed. Of these cryoforms, 78 cryoforms correspond to debris-free glaciers (Fig. 5), 17 to debris-covered glaciers, 173 to rock glaciers, 848 to gelifluction taluses and 763 to protalus lobes (Fig. 6). There is a gap in cryoforms observed between Nevado Jotabeche and El Potro Hill (Fig. 6). Around Porto Hill (app. at 4,500 m altitude), the cryoforms are found in the valleys in between the steep ridges, whereas around Nevado Jotabeche Hill and Los Tronquitos Hill, the cryoforms cover the highest points (i.e. ridges and peaks) in the region.

The gelifluction taluses cover more than 70% of the cryospheric reserves in our research area, making them the largest cryospheric reserve in the Copiapo watershed. The cryoform inventory demonstrates a notable concentration of gelifluction talus features at elevated altitudes within the southern sector of the study area, proximal to the Argentinian border (Fig. 5b, c). These talus formations overlay exposed glacier surfaces, exhibiting considerable size variability with surface areas spanning 0.01 to 20.33 km². Their spatial distribution shows strong altitudinal control, with 89% occurring above 5,000 m elevation. This elevational patterning contrasts markedly with cryoform assemblages documented in northern sectors of the research area (Fig. 6a). Gelifluction taluses appear more stand-alone, without debris-free glaciers surrounding them and only covering smaller areas of 0.01 to 3.56 $km^2$. Around Porto Hill, at 5,500 m altitude, the gelifluction taluses (2.1 $km^2$ to 20.33 $km^2$ in surface) appear in close vicinity to debris-free glaciers (0.18 $km^2$ to 6.96 $km^2$ in surface).

The second-largest cryospheric landforms in our research area are rock glaciers, making up 25.28% of the cryoforms in the region. Rock glaciers are found predominantly at high altitudes but below 5,000 m, close to steep ridges. Together with the protalus lobes, the rock glaciers are cryoforms that reach the lowest altitudes (4,200 m asl). In our research areas, Porto Hill (Fig. 6b) hosts most of the rock glaciers. They appear as elongated bodies varying between 0.02 $km^2$ and 1.2 $km^2$ that fill the valleys in between the steep ridges.

Debris-free glaciers represent the third most extensive cryospheric component within the study area, exhibiting comparable prevalence to rock glaciers. These features account for 24.58% of all documented cryoforms in the region and are predominantly located at elevations exceeding 5,500 meters above sea level (masl). The debris-free glaciers in the southern part of our research area always appear adjacent to gelifluction taluses, whereas in the north the debris-free glaciers appear as small (< 0.2 $km^2$), isolated bodies at the highest altitudes above 4,500 m. The debris-free glaciers in the south cover surface areas between 0.1 $km^2$ and 6.96 $km^2$.

In the Nevado Jotabeche Hill region (Fig. 5a), protalus lobes are the dominant cryoform. In this region, protalus lobes manifest in two distinct morphological forms: smaller features (<0.1 km²) positioned along lower slope segments and larger formations (0.1–0.4 km²) occupying gentler topographic settings. These protalus lobes are typically found at elevations predominantly below 5,000 meters..





Debris-covered glaciers are relatively rare in our research area. They are only found in the south of the Copiapo watershed (Fig. 5). They occur in front of debris-free glaciers and cover small areas of 0.05 km$^2$ and 0.1 km$^2$. They normally occur at altitudes below 5,000 m.



**Figure 5. a) Distribution map of the Farinotti et al., (2019) debris-free glacier´s thickness. b) Jotabeche volcano cryospheric reserves distribution c) The Potro hill cryospheric reserves distribution d) The Tronquitos hill cryospheric reserves distribution. . Source map provider: National Geographic, Esri, Garmin, HERE, UNEP-WCMC, USGS, NASA, ESA, METI, NRCAN, GEBCO, NOAA, increment P Corp.**





407

408

**Figure 6. a) Distribution map of the cryospheric reserves inventoried in the Atacama region García et al., (2017). b) Jotabeche volcano cryospheric reserves distribution c) The Potro hill cryospheric reserves distribution d) The Tronquitos hill cryospheric reserves distribution. . Source map provider: National Geographic, Esri, Garmin, HERE, UNEP-WCMC, USGS, NASA, ESA, METI, NRCAN, GEBCO, NOAA, increment P Corp.**

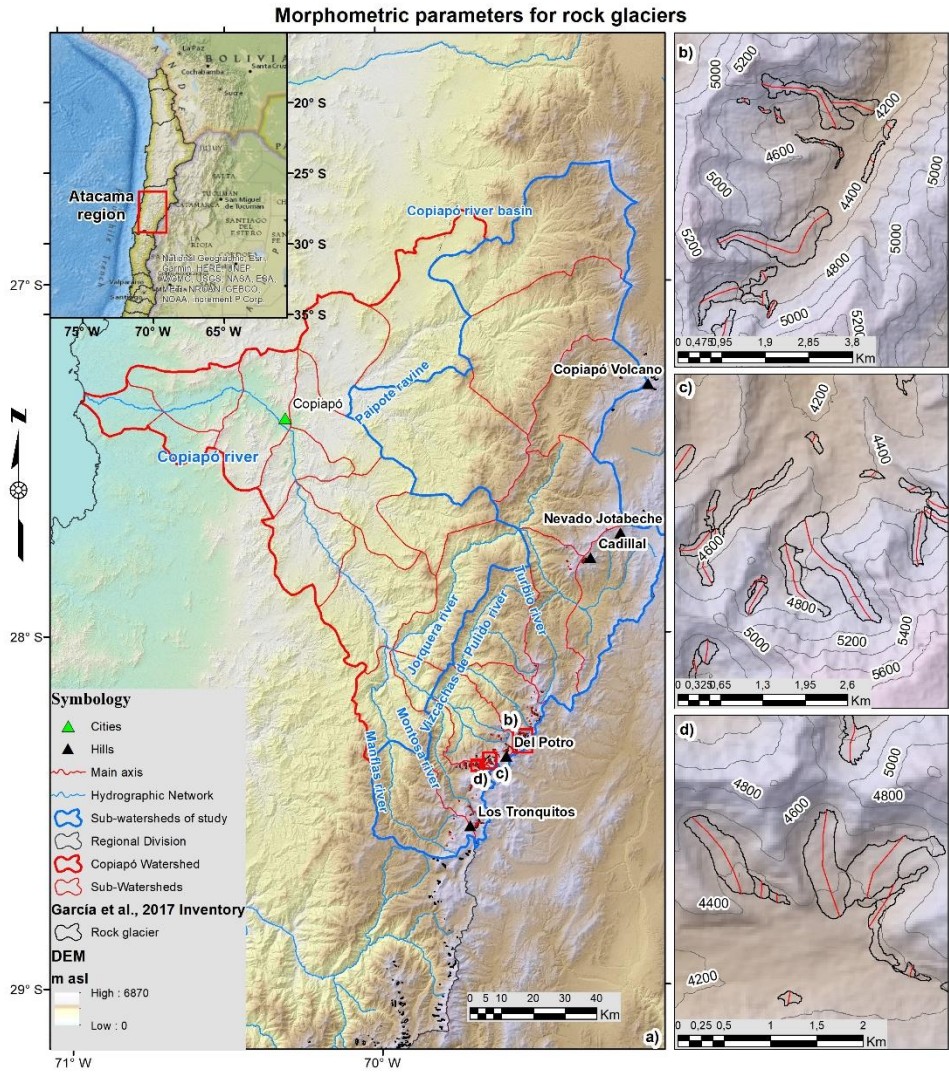

**Figure 7. a) Distribution map of the rock glaciers and main axis of Copiapó watershed. b) Rock glaciers of Ramadillas river watershed (03411), c) Rock glaciers of Del Potro river watershed (03412), d) Rock glaciers of Montosa river watershed (03413). Source map provider: National Geographic, Esri, Garmin, HERE, UNEP-WCMC, USGS, NASA, ESA, METI, NRCAN, GEBCO, NOAA, increment P Corp.**

**4.2 Snow inventory results**

By integrating all MODIS images into a global mean for the period from 2000 to 2022, we obtained the probabilities for the snow occurrence. The raster product with the calculated NCI for cryospheric resources per watershed (Fig. 8) shows that on a regional scale, the snow cover is almost permanent in the regions of high altitude above >5,500 m. These regions are located close to the Argentinian border. In the central area between the Caserones Hill and Cadillal Hill no cryoforms have been identified, which coincides with the lowest probability for snow cover, around 30% of the time. The largest snowfall anomalies





are found in the southern area of our research area. In the central area of the Copiapo watershed the snow occurrences are
concentrated around the Nevado Jotabeche Hill area. The northern part of the watershed does not show concentrated major
snowfall anomalies. Snow cover has persisted approximately 30–50% of the time exclusively at high-elevation zones (>4,500
meters above sea level) near the Argentinian border, contrasting sharply with lower altitudes starting at 2,000 m asl..
The calculated probability of the occurrence for the investigated watersheds shows that the fractional snow cover per watershed
varies, with higher snow coverage for the southern watersheds compared to the central and northern watersheds. The watershed
with the highest resource index is the Montosa river watershed (03413) with 0.18% probability of being covered in snow. The
Vizcachas de Pulido river (03410), which is located at the lowest altitude, has the lowest probability of 0.07% to have snow
coverage.

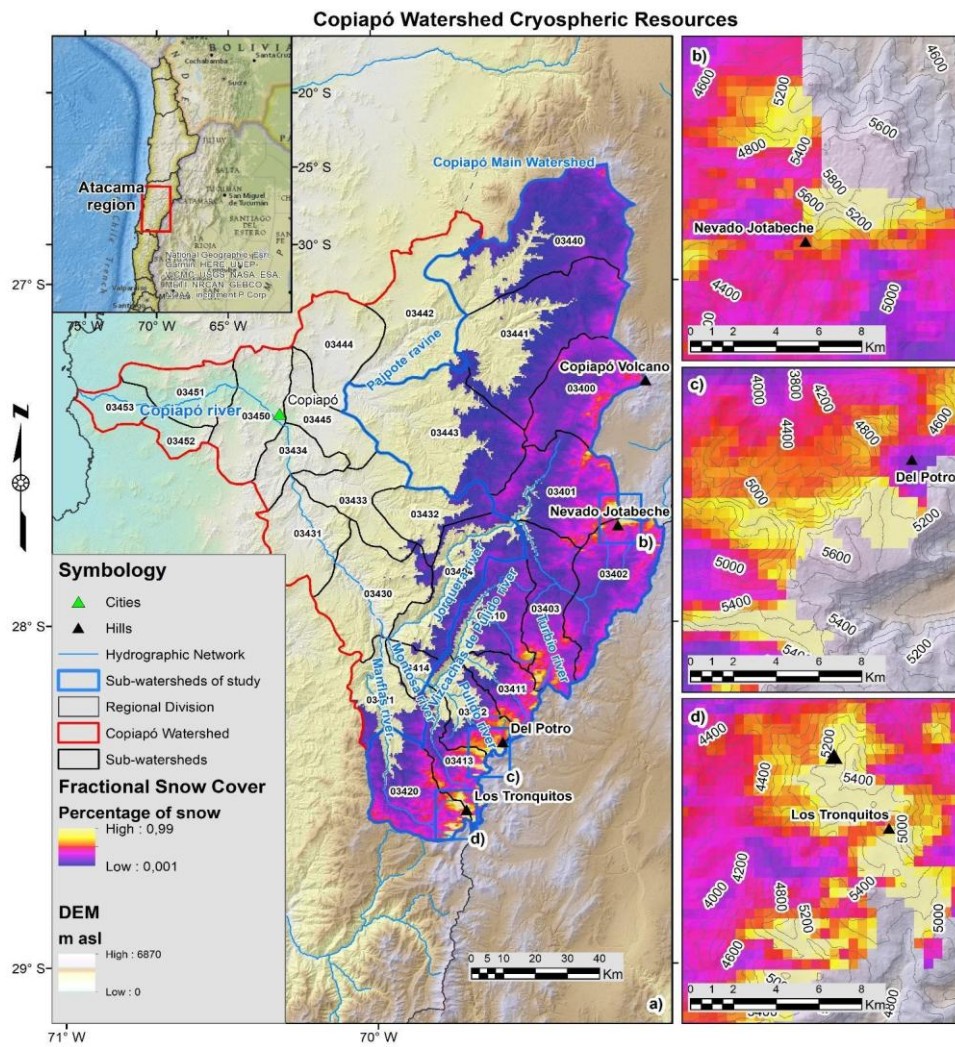

**Figure 8. a) Distribution map of the cryospheric resources in the Copiapo watershed. b) Jotabeche volcano snow permanence c) The
Potro hill snow permanece d) The Tronquitos hill snow permanence. . Source map provider: National Geographic, Esri, Garmin,
HERE, UNEP-WCMC, USGS, NASA, ESA, METI, NRCAN, GEBCO, NOAA, increment P Corp.**






### 4.3 Field survey observations of cryospheric reserves.

The fieldwork has yielded observations of cryoforms in the Del Potro and Mentosa river sub-watersheds (Fig. 9 and 10). The
Del Potro glacier (Fig. 10a) and the general periglacial environment (Fig. 9) of the Del Potro river (03412) are clearly seen in
the field. In the Montosa river the Maranceles glacier (Fig. 9b) have been observed. Other observations include well-developed
rock glaciers, gelifluction taluses, and protalus lobes in the Del Potro river (03412) (Fig. 9a, b, c and d). Glacial landforms
tend to be oriented towards areas of greater snow accumulation, generally on south-facing slopes, where insolation is lower.
In contrast, periglacial cryoforms, such as protalus lobes and gelifluction taluses, predominate on exposed slopes with greater
daily thermal oscillations. Covered and debris-free glaciers are found in depressions and high mountain watersheds, where
snow and ice accumulation is more efficient. On the other hand, protalus lobes and gelifluction taluses develop on steep slopes
and debris accumulation zones, reflecting the interaction of cold climate with steep relief. Periglacial cryoforms exhibit a
broader spatial distribution than glaciers, highlighting the role of micro-topography and local climatic variability in shaping
their formation.

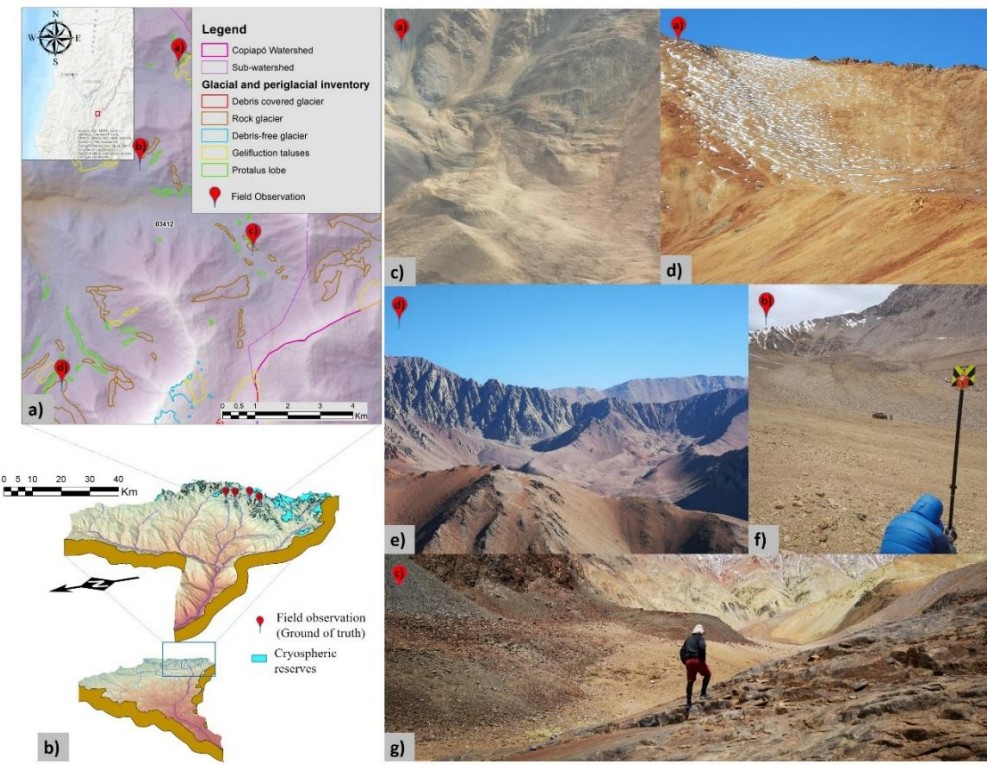


**Figure 9. Field observations of cryospheric reserves as the ground truth of the study. a) Map of cryospheric reserves distribution, b) 3D view of the watershed, c) Rock glacier of Del Potro river (03412); d) Gelifluction taluses of Del Potro river (03412); e) Rock glacier of Del Potro river (03412); f) Rock glacier of Del Potro river (03412); g) Panorama of different periglacial cryoforms of the Del Potro river (03412). . Source map provider: National Geographic, Esri, Garmin, HERE, UNEP-WCMC, USGS, NASA, ESA, METI, NRCAN, GEBCO, NOAA, increment P Corp.**





**Figure 10. Field observations of cryospheric reserves as the ground truth of the study. a) Map of cryospheric reserves distribution, b) 3D view of the watershed, c) Del Potro debris-free glacier of the Del Potro river (03412), d) Maranceles glacier of Montosa river (03413), e) Gelifluction taluses of the periglacial environment of Del Potro river (03412). Source map provider: National Geographic, Esri, Garmin, HERE, UNEP-WCMC, USGS, NASA, ESA, METI, NRCAN, GEBCO, NOAA, increment P Corp.**



### 4.4 Geophysical interpretation results

Figure 10, generated through ERT, allowed for an initial interpretation of the internal structure of the gelifluction taluses. The lobate morphology of this cryospheric feature (Figure 3a, b), along with the layout of the tomographic profile (Figure 3c, d), can be clearly observed. The tomography has a total extent of 295 m and covers a part of the body with a well-marked pattern of freezing and thawing lobulations, since 4.430 m a.s.l. upstream. This geomorphological pattern corresponds to resistivities values of 1.000 to 8.000 Ωm in the HRZ, indicating an ice rich permafrost body with a 2D extension of 20 square meters in a horizontal plane. At 4.430 m downstream, an abrupt disruption of the HRZ occurs, just in the front of the last lobulations. The LRZ begins in this area, characterized by resistivity values ranging from 50 to 200 Ωm. This anomaly likely marks the shallowest boundary of the aquifer. The geoelectric signal also indicates potential compositional variations, such as fragmented rocks within alluvial deposits.The tomography shows a direct underground geospatial connection between the ice rich permafrost and the groundwater bodies. High resistivity anomalies have enabled the estimation of an average ice core thickness of 15 meters (Fig. 11).

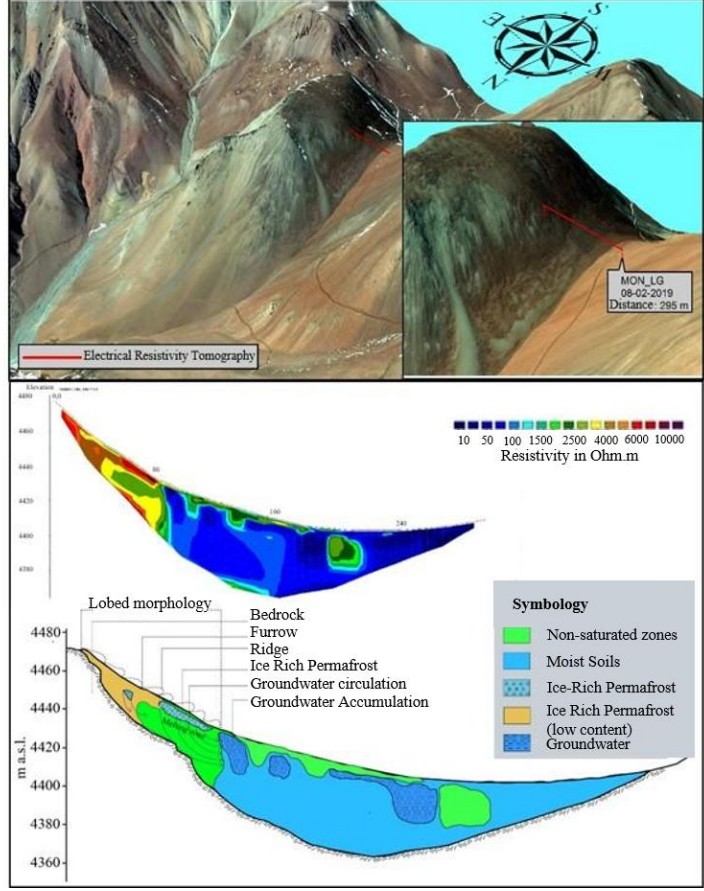

**Figure 11. Results of the electrical resistivity tomography (ERT) over the gelifluction taluses cryoforms**





An important aspect to highlight in the tomography shown in Figure 9 is that the frontal boundary of the high-resistivity
anomaly (interpreted as indicating the presence of ice-rich permafrost) aligns with the mapped extent of the cryoform,
specifically corresponding to the area where the characteristic lobe pattern of gelifluction taluses was present.

**4.5 Ice volumes, snow area and NCI for sub-watershed**

The ice volumes, snow area and the NCI for each of the sub-watersheds that constitute the cryospheric reserves of the Copiapo
river watershed have been calculated (Table 6). The watershed with the largest ice reserves is the Montosa river watershed
(03413), with 1,197 Gt, followed by the Del Potro river watershed (03412) with 0,758 Gt and thirdly the southernmost
watershed, the Manflas river watershed (03420) with 0,551 Gt of ice. In terms of volume, the largest ice volumes in the entire
watershed are found in debris-free glaciers, gelifluction taluses and rock glaciers (Fig. 12).
Following the mapping of cryospheric resources, we observe that the surface area distribution of snow cover is uneven, with
a higher density in the southern part and the northern end of the watershed. In the south, the watersheds with the largest
cryospheric resources are the Manflas river (03420), with 107.06 km$^2$ of mean snow cover (2000-2024), the Montosa river
(03413) with 75.06 km$^2$ and the Del Potro river (03412) with 53 km$^2$. In the north, the most important rivers are the Estero
Come Caballos (03402) with 122.24 km$^2$, Figueroa river (03401) with 106.72 km$^2$, Quebrada Monardes (03400) with 89.63
km$^2$ and the Cachitos river (03403) with 88.34 km$^2$ (Table 6).

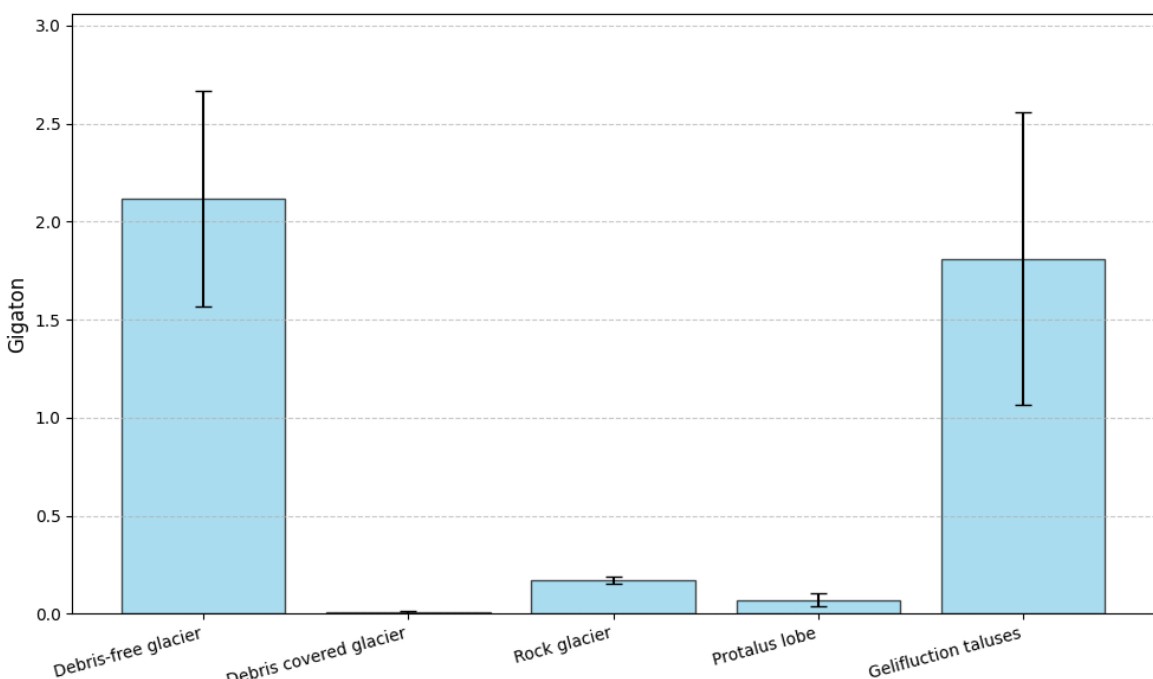

**Figure 12. Total ice volumes of the Copiapo river watershed.**





Figure 13 displays four panels that provide a comprehensive depiction of the Normalized Cryospheric Index (NCI) behavior in relation to the relative weighting of ice volume (V) and snow occurrence (S), along with the stability of sub-watershed rankings under uncertainty. Overall, the results show how the choice of weight (w) influences the mean value of the NCI and, ultimately, the final ranking of the sub-watersheds.

Panel (a) shows the evolution of the mean NCI value of each sub-watershed as w varies from zero to one. Each line represents a different sub-watershed, so that the slopes and crosses between lines allow us to appreciate which sub-watershed are favored by a greater relevance of ice volume V (when w tends to 1) or snow cover S (when w tends to 0). This preliminary behavior shows, on the one hand, the sensitivity of each subwatershed to weighting and, on the other hand, the possible dominance of one variable over the other in certain w zones.

Panel (b) shows the position in the ranking that each subwatershed occupies for each w value. The horizontal lines extend from the first position (1) to the last. It highlights consistently privileged-ranked subwatersheds regardless of *w*, in contrast to *unstably ranked* subwatersheds. This early insight into the stability of the ranking compellingly underscores the urgency of undertaking a rigorous robustness analysis.

Panel (c) presents the ranking variability function, defined as the sum (or average) of the standard deviations of the position of each sub-watershed in the different simulations. This value quantifies the overall stability of the ranking for each w. A marked minimum is seen around w=0.3, indicating that, at that point, the ranking of watersheds is less sensitive to the uncertainty of the data of V and S. In other words, assigning that weight in the combination of variables leads to a more consistent ranking in terms of robustness.

Finally, panel (d) shows the mean NCI (with its error bars) for each watershed when w=0.3, the optimal value detected with the optimization function. The vertical axis reflects the average magnitude of the index, while the error bars indicate the standard deviation derived from the Monte Carlo simulations. Thus, the visualization offers a clear identification of the watersheds with the highest NCI values, while simultaneously exposing the degree of uncertainty surrounding these estimates. This result confirms the relevance of choosing w=0.3 and provides a detailed view of the dispersion of each watershed under such a scenario.





**Figure 13. Analysis of NCI and robustness of the ranking as a function of w weighting (a) Evolution of the mean NCI for each sub-watershed as w varies between 0 and 1 (b) Position in the ranking of each sub-watershed as a function of w (c) Ranking variability function (sum of the standard deviations of the positions), where the minimum at w≈0.3 indicates the maximum stability of the ranking (d) Mean NCI ± standard deviation for w=0.3, evidencing the magnitude and dispersion of the index in each sub-watershed under the most robust condition.**

The calculated NCI, demonstrated that the sub-watershed with the highest NCI was the by the Montosa river watershed (03413) (NCI = 0.82), followed by the Manflas river watershed (03420) (NCI = 0.62), Estero Come Caballos watershed (03402) (NCI = 0.57) and Del Potro river watershed (03412) (NCI = 0.51). These are the sub-watershed with the largest cryospheric reserves. In the northern region, the Estero Come Caballos sub-watershed, it was highlighted by the index primarily attributed to its persistent snow cover.





**Table 6 Summary of results by sub-watershed of reserves in Gt, resources in km² and indexes in different scenarios.**

| Code of the sub-watershed | Reserves gigatons | Reserves uncertainty | Resources in sq. km | NCI | NCI uncertainty |
|---|---|---|---|---|---|
| 3400 | 0,057 | 0,023 | 89,63 | 0,30 | 0,04 |
| 3401 | 0,111 | 0,045 | 106,72 | 0,39 | 0,05 |
| 3402 | 0,333 | 0,128 | 122,24 | 0,57 | 0,08 |
| 3403 | 0,269 | 0,099 | 88,34 | 0,40 | 0,06 |
| 3410 | 0,025 | 0,010 | 43,90 | 0,11 | 0,02 |
| 3411 | 0,240 | 0,071 | 45,95 | 0,22 | 0,04 |
| 3412 | 0,758 | 0,199 | 53,02 | 0,51 | 0,10 |
| 3413 | 1,197 | 0,357 | 75,06 | 0,82 | 0,18 |
| 3420 | 0,551 | 0,193 | 107,06 | 0,62 | 0,11 |
| 3440 | 0,187 | 0,077 | 46,66 | 0,20 | 0,04 |
| 3441 | 0,032 | 0,013 | 46,79 | 0,12 | 0,02 |
| 3443 | 0,001 | 0,0002 | 19,76 | 0,00 | 0,01 |

## 5. Discussion

### 5.1 Cryoforms and snowfall in the Copiapo watershed

From the cryoform and snow occurrence mapping exercises (Fig. 6 and 8), we observed that the cryospheric reserves of the Copiapo watershed are predominantly located at altitudes above 4,500m. This compares well with other regions in which snowfall has been identified to be controlled by altitude (Saydi and Ding, 2020). We also observed that cryoforms occur in smaller, isolated regions in the northern part of our research area. This corresponds with our own field observations, where most cryoforms occur in smaller bodies (Fig. 6). This resulted in smaller, separated cryosphere's covering and surrounding the



highest point (i.e. field observations in Figure 6 a, b and c). They host smaller cryospheric reserves, for example near the
Cadillal Hill and the Nevado Jotabeche Hill, compared to the southern part of our study area. Because those northern
cryospheric reserves are located in or in proximity of the SAAD (Abraham et al., 2020), snowfall and permanent snow
occurrences are limited, leading to isolated cryoforms (Fig. 5). The potential volume of water that is captured in cryospheric
reserves in the sub-watersheds of the Copiapo watershed (Fig. 1) is thus strongly influenced by the South American Arid
Diagonal (SAAD) that controls the distribution of the different types of cryospheric reserves (García et al., 2017). In our case,
the proximity of the Copiapo watershed to the SAAD limits the occurrence of debris-free glaciers in the watershed north of
28.1° S latitude and limits the continuity of the cryosphere that is mainly attributed to cryoforms associated with ice-rich
permafrost, such as small rock glaciers, protalus lobes and predominantly gelifluction taluses.
In the more central domain in between Cadillal Hill and Caserones Hill no cryoforms occur. This can be explained by the
mechanical forcing of the South American low-level jet and the topographic blocking of the Andes (Insel et al., 2009). In the
central sector, Mount Pissis exhibits greater elevation on the Argentinian side. Consequently, the prevailing northeasterly
winds transporting moisture from the Pacific Ocean are obstructed by the Argentine portion of the mountain. This orographic
interception results in localized snowfall on its Argentinian slopes, rather than allowing precipitation to be deflected by the
comparatively lower-elevation Chilean topography. The topography north of Cadillal Hill and south of Caserones Hill is high
enough to block these winds on the Chilean side, leading to snowfall and eventual cryoforms in the southern regions in our
research area. A last negative spatial correlation between the snowfall and the cryoforms is the large snowfall anomaly in the
northern area. The permanent snow occurrence is high, while not many cryospheric reserves are produced. The reason for this
is the location of this region in the SAAD, resulting in melting or sublimation of the snow that does arrive in this area (Réveillet
et al., 2020), meaning that the snow does not end up in cryoforms.
From the different cryoforms, we see that gelifluction taluses cover the largest surface area in the Copiapo watershed compared
to other cryoforms in our research areas. Field observations also confirm that gelifluction taluses extend across significantly
broader areas compared to other cryoforms (Fig. 6). This is consistent with regional observations that gelifluction taluses cover
most surfaces (Garcia et al., 2017). The limited understanding of gelifluction taluses has led to their exclusion from global
cryospheric reserve inventories. Nevertheless, in assessing watershed ice volume, these taluses emerge as the most critical
features for ice storage, surpassing other formations in significance (Fig. 12). Furthermore, the ERT-derived thickness
measurements across the gelifluction lobe reveal a minimum detectable thickness of 15 meters, which markedly contrasts with
the range of 1 to 5 meters reported by Hilbich et al. (2015).
On a more regional scale, we observe that in the south of the watershed, debris-free glaciers are always in contact with
gelifluction taluses. This is confirmed in the field (Fig. 10). In the north of the Copiapo watershed, gelifluction taluses occur
more as isolated bodies. The topography of the northern area is controlled by Tertiary volcanos (Kay et al., 1994). In general,
these volcanoes exhibit gentler slopes than those of the Principal Cordillera (Harrington, 1961) and largely shape the
topography of our southern research area. Given that gelifluction taluses require steep slopes to form (Fig. 2), they are more
likely to develop in the southern sector, where the terrain is dominated by the steep relief of the Principal Cordillera unlike in





the north, where this feature is absent. Moreover, the Central Cordillera is responsible for the formation of glacial cirques
(Evans, 2006). Glacial cirques provide sheltered environments that shield snow from extreme weather conditions, facilitating
its recrystallization into exposed glaciers and other cryospheric reserves (Fig. 9d). The distribution of rock glaciers and protalus
lobes is likewise influenced by the presence of glacial cirques (Palacios et al., 2020), resulting in their predominant occurrence
in the southern portion of our study area. In contrast, no exposed glaciers are found north of Copiapó Hill (Fig. 6).Instead,
gelifluction taluses and protalus lobes allow the continuity of the cryospheric fringe of the watershed. The absence of debris-
free glaciers in this region is directly related to the South American low-level jet and the topographic blocking of the Andes
(Insel et al., 2009).
The watershed with the best balance between reserves and resources is the Montosa river watershed (03413) (Fig. 14), which
demonstrates high quantities of both cryospheric reservoirs and resources. The field observations of geomorphological features
that are indicative of active cryoforms, such as the subparallel terraces shown in Figure 10e, fit well with the mapped cryoforms
from our workflow. The mapped surface area and thickness are supported by the observed surface area and thickness in the
field. Lastly, debris-covered glaciers occur predominantly in areas with a relatively continuous cryosphere (García et al., 2017).
This is consistent with our findings that debris-covered glaciers are exclusively located in the southern region of the study
area, a zone characterized by a more extensive and contiguous cryospheric distribution relative to the fragmented northern
sectors.

**5.2 Classifying and prioritizing sub-watersheds in the Copiapo watershed**

Unlike rainwater-fed catchment systems, the Copiapó watershed operates as a cryospheric watershed. The Normalized
Cryospheric Index (NCI) identified three principal cryospheric sub-watersheds governing the hydrological dynamics of the
Copiapó system, while accounting for cryospheric reserves. These reserves were evaluated against the global permafrost model
(Gruber, 2012), exhibiting a 97% geospatial alignment with high-probability permafrost zones. The critical sub-watersheds
include: the Montosa River (03413), Manflas River (03420), and Del Potro River (03412) (Table 6). Their spatial distribution
corresponds to zones of maximum snowfall frequency and the most extensive cryospheric continuity within the Copiapó
watershed (Figures 6 and 8). These high indexes thus directly correlate with the cryoforms and snowfall occurrences. For
example, snowfall is concentrated approximately the largest debris-free glacier of the watershed, on the Cerro Del Potro river
watershed (03412) (Fig. 8c). This coincides with the location of the most important binational glacier in northern Chile, the El
Potro glacier (Garcia et al., 2017), the sub-watershed with, and the fourth highest NCI. This combination has the potential to
control the presence of water resources downstream when global temperatures continue to rise (Flores et al., 2018). With this
study, we detected the sub-watersheds that should be prioritized. In wet years, snow controls the watershed hydrology in the
northern part of the Copiapo watershed. This means that in these locations infrastructure needs to be put in place to capture
the larger amounts of run-off that will arrive with increasing warmer temperatures, such as the Estero Come Caballos sub-
watershed (03402), with the third place of the NCI. In times of water scarcity, we identified that the sub-watersheds of the




Montosa river (03413), Manflas river watershed (03420) and Del Potro river (03412) will contribute most of the flow to the
Copiapo river.
Given the observed relationship between snowfall and cryoform distribution, prioritizing the Copiapó watershed is essential
to safeguard its water resources. We propose a systematic framework to evaluate and rank sub-watersheds according to
preliminary assessments of cryospheric reserves and resource availability. From this ranking (Table 6), the main target, with
the highest NCI is the sub-watershed of Montosa river (03413). The second target is located towards the south, the Manflas
river (03420). The third target is the Estero Come Caballos watershed (3402) and finally the last target is Del Potro river
(03412) (Table 6). These four sub-watersheds constitute the basis for continuous water production in the Copiapo watershed
during dry years, because they host 83.44 % of the cryospheric reserves of the Copiapo watershed. From these four sub-
watersheds, the Montosa river watershed (03413) maintains the best balance between resources and reserves (Fig. 14). We
have shown that with this hydrological approach, the NCI allows us to infer that the sub-watershed with the highest
hydrological importance is the Montosa river watershed (03413). This index also allowed us to observe that the watersheds of
the northern region are high in cryospheric resources (i.e. snowfall and precipitation) but are less enriched in cryospheric
reserves (in the form of cryoforms). During years of substantial snowfall, management efforts should prioritize northern sub-
watersheds, such as the Estero Come Caballos sub-watershed (03402) (Fig. 14). This suggests that northern watersheds exhibit
a limited ability to develop cryogenic niches, as snow in these areas is primarily lost through melting or sublimation. A potential
explanation for the lack of cryospheric reserves in northern watersheds lies in their geomorphological characteristics, which
may hinder the accumulation and preservation of ice.
Based on the calculation of the NCI, we have been able to identify that the Estero Come Caballos watershed (03402) is the
most strategic for snow accumulation, even surpassing the watersheds in the southern part of the Copiapo watershed in terms
of cryospheric resources (Fig. 14).

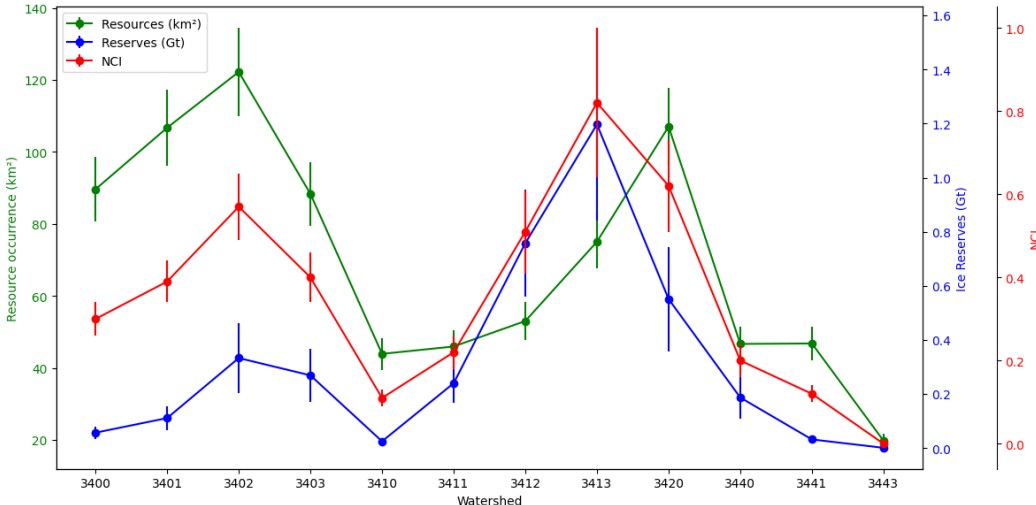


**Figure 14 Summary graph of the normalized indexes for resources, reserves and the cryosphere for each sub-watershed of the Copiapo river watershed.**



## 5.4 Impact on watershed management


We have analyzed 12 cryospheric sub-watersheds in the Copiapo watershed and showed that there are some that should be
prioritized in watershed management strategy. The methodological framework developed and presented in this study can be
extended to assess the comprehensive hydrological potential of arid watersheds in high-elevation Andean regions of Chile and
other globally distributed arid zones, with a focus on cryospheric contributions. In addition, this novel sub-watershed indexing
method is fundamental to identify those watersheds with an important snow, glacial and periglacial component that are suitable
for inclusion in the watershed protection frameworks that are implemented in the water management strategies of each
watershed in Chile. Given the persistent progression of climate change, it is imperative to integrate water reserves sourced
from cryospheric watersheds into comprehensive policy-making frameworks governing hydrological regulation.
Implementing and integrating such cryosphere inventories will also allow other regions to preserve their water resources.
Currently, the Chilean national cryoform inventory considers rock glaciers as the only cryoform of the periglacial environment.
Volumes of ice that are harbored in the other three classes of cryospheric reserves (protalus lobes, gelifluction taluses and
(debris-free glaciers) that are addressed in this study, are not considered in the inventory. Increasing the variety of cryoforms
would facilitate more accurate estimations of the volumes of ice in each sub-watershed, especially in northern Chile where the
gelifluction taluses and protalus lobes dominate the cryoforms.
Identifying the strategic position of those sub-watersheds, will ensure that the runoff remains constant, even in scenarios
without snowfall. The extraction of precious metals has consumed large quantities of water that was stored in cryoforms. A
lack of comprehensive understanding regarding the spatial distribution and volumetric potential of cryospheric water reserves
has contributed to the degradation of cryoforms critical for supplying freshwater to downstream populations residing at lower
elevations. These sub-watersheds are primarily cryospheric rather than pluvial in nature, indicating that their hydrological
input is derived exclusively from meltwater runoff originating from glaciers, periglacial ice deposits, and seasonal snowpack.
When the potential volume of water in each sub-watershed is understood, policy and infrastructure can be adapted to maintain
those cryospheric reserves and resources. Prioritizing the extraction of water from cryospheric reserves and implementing
snow capture techniques is essential to ensure the sustainable development and enhanced productivity of the examined region,
as well as the arid northern regions of Chile. These sub-watersheds may therefore play an important role in the prevention of
water disasters greater than those already known in Chile. The methodological basis developed in this study can be a key tool
for developing a law to protect Chile's cryosphere and its associated water resources when adopted in future water management
strategy plans.





## 6. Conclusions

### 6.1 Main findings of the paper

We have developed a new workflow to identify cryospheric reserves and resources and applied this method to the Copiapo watershed. The workflow consists of mapping cryoforms and snow occurrences and the calculation of a normalized cryospheric index (NCI) for sub-watersheds.

Our cryoform and snow occurrence mapping exercise resulted in the identification of cryospheric reserves in more isolated bodies north of Cadillal Hill and more continuous reserves south of Caserones Hill.

Gelifluction taluses constitute a critical strategic cryospheric reserve for safeguarding water security in northern Chile's watersheds. These landforms represent the periglacial cryoform type most likely to contain the largest ice volumes within the region's cryosphere, a hypothesis that requires further validation through targeted ERT (Electrical Resistivity Tomography) geophysical surveys of such cryospheric features.

The NCI allowed us to identify four main targets for watershed management. The four identified main targets are the sub-watersheds of the Montosa river (03413), the Manflas river (03420), the Estero come Caballos sub-watershed (03402) and the Del Potro river (03412) with NCI of 0.82, 0.62, 0.57, and 0.51 respectively. We propose that these sub-watersheds are the most important because that can sustain the runoff in the watershed in times of drought. In years with snowfall the northern sub-watersheds as the Estero Come Caballos sub-watershed (03402) have the potential to be highly productive hydraulically by snow melt. This ranking allows the prioritization of these sub-watersheds.

We conclude that the Montosa river (03413), Manflas river (03420), Estero Come Caballos (03402) and Del Potro river (03412) are the sub-watersheds that should be prioritized in the national water resources strategy because of the high NCI.

### 6.2 Limitations of this work and future research

The workflow developed in this study enables the assessment of the role of a specific cryospheric sub-watershed within a broader drainage system, providing a basis for establishing protection frameworks in water resource management. This tool is transferable to other arid regions that are sourced through cryospheric reserves and resources. However, the method is calibrated for the Copiapo river watershed, and complete resource and reserve data should be calibrated for other watersheds that require implementation of the NCI method.

To ensure more accurate error propagation in the NCI definition, it is essential to compute the uncertainty associated with the fractional snow cover product. We believe that assuming a 10% error under ideal conditions is one of the main limitations of the study of the cryospheric resources of the watershed. Our goal is to obtain a code that allows us to understand the uncertainties associated with the detection of snow in the mountainous areas of the watershed in order to improve the NCI and take it to the next level of research.



**6.3 Broader impacts**

The tool provides technical and objective data that water control agencies and those in charge of integrated watershed management, such as the DGA (General Directorate of Water), oversight boards and groundwater communities, should consider when implementing policy on the watershed management of the Copiapo watershed.

This study could help other arid regions of the planet with cryospheric reserves and resources to be able to manage the territory with a strategic plan to address the management of these cryospheric components.

**7. Open science statement**

The data generated for this project will become available upon publication. The cryospheric reserves data can be downloaded via this link: [https://doi.org/10.5281/zenodo.15633086]. The glacial and periglacial inventory for the Atacama region can be downloaded via this link: [https://doi.org/10.5281/zenodo.14921499]. The raster image for historical snow data can be downloaded via this link [https://doi.org/10.5281/zenodo.14921552].

**8. Acknowledgments**

We thank our colleagues of the Cryosphere and Water Research Laboratory for their indispensable contribution to the high altitude expeditions, and especially to project 5 of "Centro Avanzado para Tecnologías del Agua" (CAPTA) of the "Corporación de Fomento de la Producción" (CORFO) named "Tecnologías remotas y de campo para para el aumento de oferta hídrica desde fuentes superficiales y subterráneas: acumulación nival y la recarga de acuíferos", the Regional Government of Atacama and the University of Atacama, which sponsored the projects that made it possible to carry out this research and special thanks to Mr. Adrien Tavernier and Dra. Nicole Shaffer who has provided technical support in the execution of this work.

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
