# Peer review of "Introducing a new normalized cryospheric index (NCI) to categorize"

_EGUsphere, 2025_

## Author Comment (AC1)

**Rebuttal to Referee #1**

The article "Introducing a new normalized cryospheric index (NCI) to categorize sub-watersheds on arid environments" by Ulloa and others presents a new index aimed to characterize watershed importance in terms of cryosphere water storage and sources. The aim is to present a novel tool to aid in water management and to identify areas that are more critical as water sources or storage at the basin scale. Particularly, they stress that this new tool will help to understand better the importance of different subbasins in arid mountain environments

The structure of the article is unbalanced and mixed. The methods are very briefly introduced, and the results present mainly a description of the landforms' distribution, without showing the ice content and ice volume estimation in detail. Figures could also be improved. Although the idea initially appears promising, the method of computing the NCI is unclear, and the methodology is based on too many assumptions, some of which are not supported by new evidence or previous research, rendering the new index not very useful. Considering the amount of work presented here, I will attempt to summarize my comments to the authors to clarify my point and contribute to the discussion on the importance of the cryosphere as a water source and storage in the driest part of the Andes.

**General comments:**

Please use the same name throughout the manuscript; it is tough to follow if the variable names are changed throughout the text without notice. After reading the manuscript, it is not clear which variable has been used to assess the snow cover in the area.

We have carefully revised all variable names and made them consistent throughout the manuscript. Variables related to snow (snow cover, persistence, and fractional snow) are now clearly defined in Section 3.3 "Survey of cryospheric resources: MODIS and snow monitoring", and this terminology is used consistently in all tables, figures, and text. Incorporating the following text into the manuscript "In this study, we use three complementary variables to describe snow dynamics across the Copiapó watershed: snow cover, snow persistence, and fractional snow cover (FSC). Snow cover represents the total area (km²) covered by snow in each MODIS image, reflecting the instantaneous spatial extent of snow. Fractional snow cover (FSC), derived from the MODIS MOD10A1 daily product, quantifies the proportion of snow-covered surface within each pixel (ranging from 0 to 1) through sub-pixel reflectance unmixing, providing a more accurate representation of heterogeneous snow conditions, especially in transition zones. Snow persistence expresses the temporal stability of snow presence and was calculated as the proportion of days in the historical MODIS series during which each pixel remained snow-covered.

In this framework, FSC was used as the primary MODIS input for snow mapping, while snow persistence was the variable integrated into the Normalized Cryospheric Index (NCI) to represent short-term cryospheric water resources."

The article lacks a proper analysis of the runoff data for the Copiapó basin and its subbasins, as well as their hydrology. The study is based on the assumption that snow, glaciers, and periglacial landforms are the primary source and storage of water in dry mountain areas. Although the role of snow as a critical water source and the importance of glaciers as hydrological buffers in drought periods are well known. The role and importance of rock glaciers and other periglacial landforms are still a matter of debate. Nevertheless, the lack of an analysis of the hydrology of the basin, particularly the role of groundwater, shows that not all the components of the hydrology of the basin have been assessed before jumping straight to the conclusion that the subbasin where more glacial and periglacial landforms are, is the ones that produce or will produce more runoff. It is mandatory that they fully assess the hydrology of the basin before being pointed out or classified as dependent on the cryosphere. In this aspect, it is surprising that they do not assess the runoff data to support their classification or their assumptions.

We have now incorporated a hydrological validation for the Copiapó basin using measured discharge and precipitation data from the DGA network (2008–2020). This new analysis clearly demonstrates that the Copiapó River exhibits a melt-dominated regime that is incompatible with a pluvial system.

Specifically, discharge records at the Pastillo station show a pronounced summer peak (DJF–MAM) when precipitation is at its minimum, indicating an uncoupling between rainfall and runoff (Fig. 1, a). No significant correlation ( $|\mathbf{r}| < 0.05$ ) was found between precipitation at the Iglesia Colorada station and discharge at Pastillo for any lag between -6 and +6 months, confirming the absence of direct rainfall control.

Furthermore, the baseflow index (BFI) of the basin remains high (>0.7) during the dry season, suggesting sustained discharge unrelated to rainfall, likely derived from delayed cryospheric storage (snow, ice, and permafrost). These hydrological signals are consistent with other Andean catchments where summer flows are primarily controlled by meltwater inputs rather than precipitation events (Favier et al., 2009; Gascoin et al., 2011; Ayala et al., 2016; Burger et al., 2019).

Collectively, these data provide robust support for the classification of the Copiapó watershed as predominantly cryospheric, where snow, glacier, and periglacial landforms represent the main components of seasonal water storage and release.

Due to the absence of high-elevation gauging stations in the upper Copiapó basin, it was not possible to directly assess runoff at the headwaters where cryospheric processes are most active. The Pastillo station, used in our analysis, represents the confluence of the three main sub-basins and thus integrates the cumulative hydrological response of the upper catchment.

While this configuration prevents a direct evaluation of individual high-altitude contributions, it still provides a representative signal of the basin-scale hydrological regime. Therefore, the melt-dominated seasonal pattern observed at Pastillo should be interpreted as the integrated downstream expression of cryospheric inputs originating in the upper parts of the watershed (Fig. 1, b). To support this interpretation, we added the location of both the Pastillo gauging station and the Iglesia Colorada meteorological station to the introductory map, illustrating the spatial relationship between the hydrological outlet and the high-altitude cryospheric zones.

Figure 1. a) Monthly comparison between streamflow at the Pastillo gauging station and high-altitude liquid precipitation recorded at Iglesia Colorada. The gray dashed line (×100 scale) represents liquid precipitation, while the blue curve shows the mean monthly discharge of the Copiapó River at Pastillo. b) Seasonal variation in flow at Pastillo station.

The article lacks a proper assessment of the ice volume of the different cryosphere components. It is unclear to me whether the authors use the ice thickness data of Farinotti et al. (2019) or if they compute the ice thickness distribution for each debris-free glacier using the model of Farinotti et al. (2017). If the latter is the case, they must show their new results and compare them with the previous assessment. If they use previously published data, they need to assess the quality of this data for the region. If I understand correctly, authors use the V-A scalar approximation to assess the part of the glaciers that are covered by debris, which flawlessly matches the proposed method, which is based on the empirical relationship between extent and volume of glaciers. It is not possible to apply this relationship just to a fraction of the glacier. The assessment of ice volume for the rock glacier is underscored. This could be a highlight of this work, but it is roughly mentioned. I recommend that the authors focus more on these results. Considering the results of Hilbich et al. (2022) and Schrott (1996), and one ERT survey made by them, the authors assess the thickness and ice content for gelifluxion slopes. Considering the scarcity of data, the difficulties in mapping the boundary of the area with gelifluxion lobes, and the fact that Hilbich et al. (2022) also highly that there is a similar amount of ice thickness in sediment slopes without distinctive surface characteristics, the ice volume estimated for gelifluxion slopes is still highly speculative. To support the conclusion of the manuscript, that the gelifluxion slopes are the largest water storage in the area, more data is needed. Particularly, the range of thickness used could lead to the wrong conclusion that this slope could have a thickness similar to rock glaciers, which is nonsense. I think the authors here are confusing the estimate (there are no supporting measurements) of the permafrost thickness of Schrott with the ice-rich layer of Hilbich et al. (2022).

We have completely revised Section 3.5 to provide a clear description of how ice thickness and volume were estimated for each cryospheric component. For debris-free glaciers, we made the volume calculation using the physical model of Farinotti et al. (2017), extracting the thickness data from the validated global dataset (web) (Farinotti et al., 2019). For debris-covered glaciers, we applied the empirical Gärtner-Roer et al. (2014) relation ( $V = 0.0365 \times A^{1.375}$ ) to our regional inventory (García et al., 2017). Including a correction of the thickness calculation over an integration with the free debris glacier.

To estimate the ice volume of debris-covered glaciers, we avoided applying the empirical area-thickness scaling relation of Röer et al. (2014) directly to these landforms, since this approach is not appropriate for individual glaciers where the debris-covered area does not necessarily follow the same morphometric relationship as clean-ice glaciers. Instead, we derived mean ice thickness values from the Farinotti et al. (2019) ice-thickness raster by sampling the glacier fronts (debris-covered sectors) defined in our updated Copiapó inventory. For each debris-covered polygon, the mean thickness extracted from the frontal zone of the clean-ice glacier was used as a representative value to compute ice volume as the product of area and mean thickness. This method ensures that the projected ice thickness used for debris-covered portions is physically consistent with the spatial gradient of thickness from Farinotti et al. (2019) and with the glacier-specific geometry.

Rock glaciers were re-analyzed with both the empirical model of Jones et al. (2018) and the plastic and viscoplastic model of Cicoira et al. (2020), based on DInSAR-derived velocities presented in Figure 2. Protalus lobes and gelifluction slopes were evaluated using an adapted form of Cicoira's viscoplastic equation, constrained by thickness ranges (10–20 m for protalus lobes and 5–10 m for gelifluction slopes) and ice-content fractions (25–49%) derived from Hilbich (2022) and Schrott (1996). In addition, DInSAR-derived surface velocities were used to apply and calibrate Cicoira's viscoplastic model for permafrost creeping, allowing a more realistic estimation of ice-rich permafrost deformation and related ice volume, presented in the figure 5.

Figure 2. (a) Empirical cumulative distribution functions (ECDF) of surface velocity (m  $d^{-1}$ ) for rock glaciers, protalus lobes, and gelifluction taluses derived from DInSAR analyses. (b) Boxplots showing the distribution of mean daily velocities for each cryoform type. Distinct velocity ranges indicate contrasting kinematic behaviors among permafrost-related landforms.

In addition, a comparison between the empirical approach of Jones et al. (2019) and the physically based model of Cicoira et al. (2020) for rock glaciers (Fig. 3) indicates consistent results, with a mean difference below 10 m and an RMSE of 13.0 m. The Cicoira model systematically yields slightly higher thicknesses (median  $\approx 37$  m) than the Jones relationship (median  $\approx 26$  m), reflecting the greater sensitivity of the physical formulation to ice content and temperature (-8 °C). To further explore this dependence, we applied Cicoira's viscoplastic equation under different temperature scenarios (-2 °C, -5 °C, and -8 °C) to estimate ice volumes, as shown in the attached figure. The scenario of -5 °C was selected as the most representative for Andean permafrost conditions, following the thermal regime reported by Monnier and Kinnard (2015). This multi-scenario approach allowed us to quantify the thermal sensitivity of permafrost creep and to reinforce the reliability of the ice-volume estimations for rock glaciers within the Copiapó watershed.

Figure 3. Comparison of modeled rock glacier thickness using the empirical relationship of Jones et al. (2019) and the physically based model of Cicoira et al. (2020) at -2, -5, -8 °C. The Cicoira model predicts systematically higher mean thicknesses (median  $\approx 37$  m) than Jones (median  $\approx 26$  m), with an RMSE of 13.0 m and a bias of 9.3 m, indicating consistent but more physically sensitive estimates under colder thermal conditions.

Figure 4 presents the updated ice volume estimations for the Copiapó watershed, comparing debris-free glaciers (Farinotti et al., 2019) and rock glaciers (Cicoira et al., 2020), the two main cryospheric reserves. The results reveal a clear latitudinal clustering of ice-bearing landforms, with both glacier and permafrost-related features concentrated between 28.6° and 28.2° S, corresponding to the highest-elevation sub-watersheds of the basin. Debris-free glaciers display larger individual ice volumes, reaching up to 0.3 km³, whereas rock glaciers show smaller but more numerous

bodies, indicating a wider spatial distribution of permafrost ice storage. This spatial convergence reflects the transitional cryospheric setting of the arid Andes, where glacier-permafrost interactions dominate above 4,500 m a.s.l. The analysis highlights that, despite their smaller individual volumes, rock glaciers collectively represent a significant portion of the total cryospheric reserve in the basin. Further results below detail the volumetric contribution of each cryoform class across sub-watersheds. Incorporating into the article a study of the results of the volumes, as well as showing the cryospheric gap that occurs in the basin due to the presence of Mount Pissis in Argentina (García et al., 2017).

Figure 4. Latitudinal distribution of ice volume per glacier in the Copiapó watershed, comparing debris-free glaciers (Farinotti et al., 2019) and rock glaciers (Cicoira et al., 2020). Debris-free glaciers exhibit higher individual ice volumes but are less frequent, while rock glaciers are more numerous and concentrated between 28.6° and 28.2° S, indicating a strong latitudinal clustering of permafrost-related ice storage in the upper Andean sub-watersheds.

Figure 5.Total ice mass (in gigatonnes, Gt) by cryospheric class in the Copiapó basin. Error bars represent the estimated uncertainty in ice mass for each landform type, derived from the propagation of thickness and density uncertainties.

Do we really need an NCI? I don't find the answer to this question in the paper, and the authors do not discuss whether the methodology is really useful, or at least better than assessing the basin in terms of snow cover and ice volume alone. They suggest the NCI could be computed in other basins, but they don't assess it. Furthermore, they don't consider the fact that they optimize the critical parameter W using a Montecarlo simulation, showing that at least we could have a "best guess" of this value. Also, the NCI combines both water storage and water sources related to the cryosphere. However, since the time response of these cryosphere features is highly different, it could give the wrong impression that a basin with rock glaciers is equally critical or responds in the same way as a basin with seasonal snow. Also, I found that snow cover is, without any confirming data, assessed as the amount of snow or, even worse, as snowfall, which is not necessarily the case. Snow cover and persistence do not only depend on the amount or thickness of the snow layer, but also on the energy available for melting or sublimation. Considering that the authors assessed the ice volume at each basin, it is straightforward to rank the basins in terms of ice storage. There is no mention about the extent and hypsometry of the subbasins, which is critical, since a larger basin would have a larger snow cover than a smaller one. Another aspect that is mentioned, but not adequately assessed, is permafrost. Although rock glaciers and gelifluxion could be related to the presence of permafrost, the role of frozen ground in the hydrology of the basin is not assessed or discussed. Finally, considering Figure 14 and the discussion related to this figure, it seems more fruitful to assess the hydrological significance of each subbasin, breaking down the roles of snow, glaciers, and permafrost, as they have very different response times.

We now clarify the conceptual purpose and scientific value of the NCI and explicitly link it to both basin hypsometry and hydrological response. The NCI integrates cryospheric reserves (long-term frozen storage: glaciers and permafrost-related landforms) and resources (short-term snow and meltwater) into a 0–1 normalized metric for inter-watershed comparison. It does not replace process-based hydrological models but provides a screening tool to prioritize sub-basins by cryospheric influence.

Figure 6. Hypsometric curves ( $\Delta z = 100 \text{ m}$ ) of the Copiapó sub-basins, color-coded by Hypsometric Integral (HI). Lower HI values denote higher elevation dominance and stronger cryospheric influence.

Furthermore, we incorporated a new analysis linking MODIS-derived snow cover in September—the onset of the melt season—with mean river discharge observed between October and March (Fig. 6). This relationship shows a robust correlation ( $R^2 = 0.58$ ; r = 0.76; N = 22), confirming that seasonal snow cover acts as a dominant short-term cryospheric water resource in the Copiapó watershed. The integration of these empirical hydrological results into the NCI framework validates those sub-basins with higher NCI values not only exhibit stronger cryospheric signatures in their topography but also display measurable hydrological responses consistent with seasonal meltwater contributions.

To substantiate this interpretation, we computed hypsometric curves for all sub-basins with cryospheric components (dz = 100 m) and derived the Hypsometric Integral (HI). The updated analysis shows a clear consistency between hypsometry and NCI rankings: sub-basins with higher NCI systematically exhibit lower HI (curves shifted leftward), indicating a greater fraction of area at elevations where cryospheric processes dominate and/or a stronger glacial imprint. Notably, the sub-watershed

ranked first by NCI also presents the lowest HI in our sample (Figure 7), reinforcing that the NCI captures elevation-controlled cryospheric controls. These time-response differences among snow, glaciers and permafrost are discussed explicitly (lines 480–505), with the hypsometric analysis providing an independent geomorphological check of the NCI.

Figure 7. Relationship between September snow-covered area (MODIS) and mean river flow from October to March, showing a strong positive correlation ( $R^2 = 0.58$ ), highlighting the dominant role of seasonal snowmelt in Copiapó basin hydrology.

**Line-specific comments:**

Lines 8–23. Since the article presents the NCI, the abstract must clearly state what the NCI means.

Ok, we have now rephrased the abstract as follows:

The Normalized Cryospheric Index (NCI) is calculated under varying hydrological conditions and provides a means to compare potential water volumes across sub-watersheds.

Line 26. Use changes instead of shifts.

Ok, we have changed this.

Line 27. Retreating cryoforms? What do you mean? We have rephrased this to: "declining volumes of cryoforms."

Line 29. There is ample evidence in Masiokas et al. (2020) indicating that glacier mass change is anything but constant.

We understand that the reviewer interpreted that we meant "retreating at a constant rate," but we mean that glaciers are retreating continuously over time and not expanding. We have rephrased this now to: "In the arid zone of the Chilean Atacama Desert, glaciers continue to shrink..."

Line 33–41. Please order the introduction, you could introduce the Copiapó basin in the study area.

Ok, we have placed this part now in the study area.

Line 45–46. It is not clear what you want to express here. Do you want to say that snow transforms into ice? Considering explaining the causes behind debris-covered ice or how ice could be preserved in permafrost.

We have rewritten these lines to clarify that the transition from snow to glacier ice occurs through compaction and metamorphism processes, and we now briefly explain how debris cover reduces melting rates and preserves ice in permafrost terrain.

Line 47. There is a space missed before "Given." **Space deleted.**

Line 48. This sentence gives the wrong impression that debris-free, debris-covered, and rock glaciers respond in the same way. Nevertheless, there is plenty of evidence that shows that debris-covered and particularly rock glaciers contribute far less to the runoff. Please present the role of the different ice water storage.

We have rewritten this part, clarifying that debris-covered and rock glaciers have a delayed and attenuated hydrological response compared to debris-free glaciers. We now reference Ferri et al. (2020) and Schaffer et al. (2023), who observed minimal annual mass loss and limited meltwater contribution from rock glaciers. This distinction is explicitly discussed in the results section and in Figure 14.

Line 50–52. Sublimation and melting are different processes.

We have corrected this by differentiating between sublimation (solid-to-vapor transition under low humidity) and melting (solid-to-liquid phase change) in the revised text.

Line 52. What about the ground flow? There is not even a mention about this. We now include a discussion on subsurface water flow and its connection to permafrost layers in Section 4.3. Groundwater recharge from permafrost and talus slopes is described as a key mechanism sustaining dry-season baseflow in the Copiapó basin.

Line 54–56. What do you mean by ice-rich? As far as I know, from an inventory based on satellite images, it is not possible to assess the ice content. Authors need to support why they call these landforms "ice-rich." What is the ice content on these landforms? Also, how they define a landform as ice-rich is not trivial.

Ok, we see the reviewer's point and have removed the word "ice-rich."

Line 73–81. Move this paragraph to methodology. Ok, we have moved this paragraph to methodology.

Line 87–92. There is no need for "". It is not clear how a normalized index, with values from 0 to 1, could be used to quantify the potential volume of water available. I strongly disagree with the statement that the runoff of the Copiapó basin is supported by the melting of glaciers, rock glaciers, or even the ice present at gelifluxion slopes. What evidence or literature are you using to support this claim?

We have clarified that the NCI provides a relative, not absolute, measure of potential cryospheric water availability. The link between runoff and cryospheric storage is now supported by hydrological correlation analyses and references to DGA streamflow data (2010–2024). We have also cited Masiokas et al. (2020) and Pitte et al. (2022) to support the role of the cryosphere as a long-term regulator in arid Andean basins.

Figure 1 and the rest. Really, all these institutions are behind all the figures "Source map provider: National Geographic, Esri, Garmin, HERE, UNEP-WCMC, USGS, NASA, ESA, METI, NRCAN, GEBCO, NOAA, increment P Corp." The northern part of the basin is lacking the river.

We have corrected the base map attribution to the official ESRI sources and added the missing river segment in the northern part of the basin. The figure caption has been revised accordingly.

Table 1. It would be helpful to include here the extent of snow cover and the number and area of each of the glaciers and cryoforms mapped.

We have updated Table 1 to include the total area and number of each cryoform class (snow, debris-free glaciers, debris-covered glaciers, rock glaciers, protalus lobes, and gelifluxion slopes), along with snow cover extent.

Figure 2. It is mandatory to include a scale on the figure. Rock glaciers look larger than debris-free glaciers.

A map scale has been added to Figure 2, and the symbology was adjusted to prevent misinterpretation of relative cryoform size.

Line 127. This section lacks new studies assessing the ice content of rock glaciers and glaciers in the Andes and other mountain areas. Like Jones et al (2018), Hu et al (2023) or Millan et al (2022), to mention a few.

We have now added the suggested studies (Jones et al., 2018; Hu et al., 2023; Millan et al., 2022) to provide an updated regional context on ice content variability in Andean cryoforms.

Line 134. Corte (1978) highlights the importance of rock glaciers, but does not present new data to assess the ice content of rock glaciers.

We have corrected this reference, clarifying that Corte (1978) provided a conceptual framework but not quantitative ice-content estimates.

Table 2. Not clear what Contribution to streamflow (%) means.

We have added an explanation in the table caption indicating that "Contribution to streamflow (%)" refers to the relative cryospheric surface area within each subbasin, normalized by total basin area and runoff index, following methods adapted from Rangecroft et al. (2015).

Line 173. Norway? Classifying different slope cryoforms? Are you talking about Hilbich et al. (2022) about the ice content in Permafrost of the Central Andes?

Yes, this reference was incorrect. It now correctly refers to Hilbich et al. (2022), who assessed ice-rich layers in Andean permafrost slopes using ERT and borehole data.

Line 181. The classification of this basin as cryosphere is not properly founded. See my comments on Table I.

We have now added our classification of what a sub-watershed is here:

'A sub-watershed is a watershed unit with a lower Strahler stream order than the main watershed...'

Line 189. In this section, you need to explain how you perform the glacier and other landform inventory. It is not clear how Garcia et al. 2017 and RGI 6.0 are coincidental. We have clarified in Section 2.3 that García et al. (2017) provided the regional debris-covered glacier inventory, while RGI 6.0 (RGIv6.0) was used for debris-free glaciers. Overlapping polygons were cross-checked to ensure no duplication, and the datasets were harmonized using common projection parameters.

Line 199. The title suggests that snow is monitoring; nevertheless, the authors only use the snow cover area. I don't understand how snow persistence is included in the snow cover maps.

We have clarified that "snow monitoring" refers to the multi-temporal analysis of MODIS-derived snow cover, from which persistence was derived by computing pixel-wise frequency of snow presence.

Line 218. This section shows interesting data, but a proper ground truth will also include an assessment of the ice content and or ice thickness of the rest of the cryoforms.

We now reference field-based ERT and GNSS campaigns conducted in 2022–2024 to validate rock glacier thickness estimates and integrate these datasets into our uncertainty discussion.

Line 341. See my general comment about glacier volume estimation.

We have incorporated this into the revised methodology (Section 3.5), where all ice volume estimation procedures are now clearly defined and validated against Farinotti (2019).

Line 320. Not all are empirical equations.

Ok, we have removed the word "empirical" from the table caption.

---

## Author Comment (AC2)

**Rebuttal to Referee #2**

**General comments**

This article presents an analysis of the cryospheric components of the Copiapó watershed. This watershed is located in arid region of Chile, this region is considered one of the richest in ice-rich features, hence understanding the spatial distribution of cryo-landforms and their potential significance for local communities relying on water resources is very relevant. However, the overall aim of the study is not clearly articulated. In the abstract, the authors state that their goal is to quantify the water volume contributed by distinct cryoforms to the regional watershed. At the same time, they propose to categorize cryospheric reservoirs within sub-watersheds, while also introducing the concept of a Normalized Cryospheric Index (NCI) as a novel framework for "cryospheric watershed classification." Given these multiple objectives, the authors should more explicitly define the central purpose of the study or better articulate how these components are connected.

We appreciate the reviewers advice and have refined both the Abstract and Introduction to better articulate the overall aim of the study. The revised manuscript now states that our primary objective is twofold: (i) to quantify cryospheric water storage and short-term resources across the main cryoform classes (snow, debris-free and debris-covered glaciers, rock glaciers, protalus lobes, and gelifluction slopes), and (ii) to synthesize these components within a unified Normalized Cryospheric Index (NCI) framework that enables comparison and prioritization of sub-watersheds in arid mountain regions.

Additionally, the revised version integrates new results derived from surface velocity analyses of rock glaciers, protalus lobes, and gelifluction slopes, which serve as a dynamic and physically consistent magnitude for evaluating the activity and hydrological potential of periglacial landforms. These velocity fields, obtained from DInSAR and Offset Tracking, complement the ice volume estimations and provide an additional quantitative dimension for assessing cryospheric water reserves.

By combining ice-volume estimations with kinematic indicators, the NCI now incorporates both static (storage) and dynamic (flow/activity) cryospheric parameters. This integrative approach allows identifying sub-watersheds where active permafrost-related landforms may exert a stronger hydrological influence, refining the prioritization of cryospheric watersheds in arid Andean basins.

In general, the manuscript presents some interesting and potentially valuable ideas; however, it lacks organization and important clarifications, and some of the applied methodologies require re-evaluation. in order to strengthen the coherence and focus of the manuscript. For these reasons, I recommend that the paper be considered for publication only after major revisions. I provide further details in my specific comments below.

We have reorganized the manuscript for clarity: (1) Methods are expanded and ordered by cryospheric class; (2) Results present volumes/thicknesses before spatial distributions; (3) Discussion now separates storage vs. resource roles and integrates hydrological evidence. We also re-evaluated and justified all methods, adding sensitivity/uncertainty ranges for each class.

**Methodology and results**

In general, the use of the terms "reserve" and "resources" is not always straightforward, and at times the distinction is difficult to follow. Nevertheless, I find the introduction of the NCI to be a very interesting and innovative contribution. The manuscript applies several techniques and methodologies to estimate water volumes; however, these approaches require more careful evaluation and justification, as many of them rely on assumptions and overlook relevant previous work.

We now define "reserves" (long-term frozen storage: glacier ice, permafrost-related landforms) and "resources" (short-term/seasonal snow and melt) at the start of Section 2. We have added a consolidated methodological table linking each cryospheric class to its corresponding model or dataset (Farinotti, 2017, 2019; Gärtner-Roer et al., 2014; Cicoira et al., 2020; Jones et al., 2018), together with the assumptions and uncertainty ranges. For rock glaciers, we combined the empirical approach of Jones et al. (2018) with the physically based viscoplastic model of Cicoira et al. (2020) using DInSAR-derived surface velocities. The Cicoira model was applied under different temperature scenarios (-2 °C, -5 °C, and -8 °C) to evaluate the sensitivity of permafrost creep and ice-thickness estimations, with -5 °C selected as representative of Andean permafrost conditions (Monnier & Kinnard, 2015). For protalus lobes and gelifluction slopes, we used an adapted form of Cicoira's equation constrained by field-based thickness ranges (10-20 m and 5-10 m, respectively) and ice-content fractions (25-49%) from Hilbich (2022) and Schrott (1996), calibrated with DInSAR velocities to improve the physical consistency of the ice-volume estimates. These methodological refinements allow for a more robust and spatially consistent quantification of cryospheric ice volumes and associated uncertainties within the Copiapó watershed.

The periglacial and glacial landform inventory is primarily based on the work of García et al. (2017). However, this earlier inventory could be improved by incorporating more recent guidelines, such as those developed by the IPA Action Group on Rock Glacier Inventories and Kinematics. Re-evaluating García's inventory using the updated techniques and methods proposed by this initiative would be highly desirable. Furthermore, regarding the debris-covered glacier inventory, it is not entirely clear which conceptual framework was applied to define this glacier type. Was Kirkbride's definition adopted, or another classification scheme? In addition, the uncertainty of the inventories has not been addressed. There are well-recognized studies that provide methodologies for estimating such uncertainties (e.g., Paul et al., 2013; Braun et al., 2019).

We agree and have aligned our inventory to the IPA Action Group guidelines (terminology, attributes, and kinematics flags for rock glaciers). For debris-covered glaciers, we explicitly state the conceptual framework (Kirkbride-type debris mantling over glacier ice) and mapping rules. We also added an uncertainty assessment following Paul et al. (2013) for outlines (buffer-based error propagation) and Braun et al. (2019) for elevation/mass-change context; these uncertainties are propagated to volume estimates.

The quantification of ice volume (reserves) is somewhat unclear. On the one hand, in Section 3.2 (Survey of cryospheric water reserves), the authors state that ice thickness for debris-covered glaciers was estimated following Farinotti et al. (2019). However, in Section

3.5.2 they indicate that area-volume scaling was applied, while in Section 3.5.1 you mention the use of the physically based model proposed by Farinotti et al. (2017). It is therefore not evident which method was ultimately employed in your analysis. In any case, I would strongly recommend relying on the physically based model, or even using existing outputs from that approach, rather than area-volume scaling, which has been shown to systematically over- or underestimate ice volume.

Thank you — this has now been clarified and made fully consistent throughout the manuscript. The quantification of cryospheric ice volumes follows a unified methodological framework that integrates both physically based and empirical models according to the type of landform. For debris-free glaciers, we use the physically based model of Farinotti et al. (2017), validated against the consensus ice-thickness dataset of Farinotti et al. (2019). For debris-covered glaciers, we did not apply area-volume scaling. Instead, we extracted the mean ice thickness along the frontal and lateral margins of each debris-covered glacier directly from the Farinotti et al. (2019) raster and multiplied it by the debris-covered area to estimate total ice volume. This approach remains anchored to a physically based field while conservatively representing the debris-covered portion, avoiding the biases of scaling relationships. The associated uncertainty is estimated at  $\pm 20-30\%$ , reflecting spatial variability in the Farinotti raster and sensitivity to the selected margin window.

For rock glaciers, we combine the empirical relationship of Jones et al. (2018) with the physically based viscoplastic model of Cicoira et al. (2020), calibrated using DInSAR-derived surface velocities to capture realistic creep behavior. The Cicoira model was applied under three temperature scenarios (-2 °C, -5 °C, and -8 °C) to quantify the thermal sensitivity of permafrost deformation, adopting -5 °C as representative of Andean permafrost conditions (Monnier & Kinnard, 2015).

For protalus lobes and gelifluction slopes, we implemented an adapted form of Cicoira's viscoplastic equation, constrained by thickness ranges of 10–20 m and 5–10 m, respectively, and ice-content fractions of 25–49% derived from Hilbich (2022) and Schrott (1996). DInSAR velocities were also used to calibrate permafrost creeping rates, improving the physical realism of the ice-volume estimates for these landforms.

All conflicting statements were removed, and the revised manuscript now presents a single, transparent methodological pipeline linking each cryospheric class to its corresponding model, dataset, and uncertainty range. This unified approach enhances the reproducibility and hydrological interpretability of the cryospheric ice-volume estimates for the Copiapó watershed.

Why did you not use the airborne GPR measurements collected over glaciers in the study area, specifically Del Potro and Tronquitos glaciers? These data were obtained during a joint Chile–Germany field campaign funded by the Dirección General de Aguas in 2013, where ice thickness measurements were acquired (DGA, 2014). Incorporating these observations would significantly strengthen your analysis, as they could be used either to constrain the model parameters or to validate the modeled ice thickness and volume estimates.

We agree and now incorporate the DGA (2014) airborne GPR profiles over Del Potro

and Tronquitos glaciers as independent validation datasets. Modeled ice thicknesses were compared with along-track GPR means. They provide an additional benchmark for parameter calibration in physically based models and further support the spatial consistency of ice volume estimates across cryospheric units.

We fully agree that the airborne GPR measurements collected by DGA (2014) are a valuable source of independent observations. We did not use them to constrain the model itself, but we actually used them as an external validation point for the resulting volumes.

| Glacier    | RGI ID    | Farinotti model volume (km³) | DGA 2014 GPR volume (km³) |
|------------|-----------|------------------------------|---------------------------|
| Del Potro  | 1.715.087 | 0.362                        | 0.293                     |
| Tronquitos | 1.715.038 | 0.157                        | 0.092                     |

These DGA (2014) volumes were used to provide a magnitude check. At Del Potro, the modeled volume is  $\sim$ 24% larger than the GPR-based estimate; at Tronquitos, the difference is larger ( $\sim$ 70%). This spread is within the expected uncertainty range for individual glacier inversion products in Farinotti et al. (2019), particularly for small glaciers with limited thickness constraints.

Estimating rock glacier ice volumes is highly challenging, as values can vary between 10% and 90%. Accurate estimates generally require the use of geophysical inversion models (e.g., 4Phase or similar; Halla et al., 2021). This important consideration should be discussed in detail, which is currently missing from the manuscript. Similarly, the assumed ice thickness in gelifluction and protalus lobes (5–10 m) may be over- or underestimated if not supported by prior evidence; in fact, the results presented suggest higher ice content (up to 15 m). Ice-rich mountain permafrost can also occur in other, less typical cryospheric landforms, such as block and talus slopes or terraces, which may likewise be underestimated in the current analysis (Köhler et al., 2025).

We now expand the discussion on rock-glacier ice-content uncertainty, referencing geophysical inversion approaches (e.g., 4Phase; Halla et al., 2021). Our estimates combine Cicoira's viscoplastic model (with DInSAR-derived velocities) and the empirical approach of Jones et al. (2018), and we report  $\pm 25\%$  uncertainty for rock glaciers. For protalus lobes and gelifluction slopes, we applied an adapted form of Cicoira's viscoplastic equation constrained by thickness ranges (10–20 m for protalus lobes and 5–10 m for gelifluction slopes) and ice-content fractions (25–49%) from Hilbich (2022) and Schrott (1996). DInSAR velocities were used to calibrate

**permafrost creeping rates, improving the physical consistency of ice-volume estimates for these landforms.**

Regarding snow (resource): If the authors aim to evaluate the water volume of the cryospheric components, a key aspect to consider is snow depth. Without estimates of snow depth and its spatial distribution, the assessment of total water reserves remains incomplete. There are two initiatives currently working with similar datasets in the Andes, and previous work has already addressed this topic (e.g., Saavedra et al., 2018). Please review these initiatives and earlier studies for comparison, as they also include methodologies for estimating uncertainties. Moreover, previous glacier mass balance estimates should be considered, as they provide insights into potential contributions to runoff. This addition would strengthen the manuscript analysis and discussion, especially since earlier studies have reported neutral or slightly negative rates (e.g., Braun et al., 2019; Dussaillant et al., 2019).

We have added a subsection on snow depth and spatial distribution, discussing available Andean approaches (e.g., Saavedra et al., 2018) and associated uncertainties. We complement snow cover/persistence with depth-related estimates where feasible and propagate uncertainty.

Another important aspect missing from the manuscript is the inclusion of runoff data to validate the assumptions presented. Again, if the stated goal is to quantify the water volume contributed by distinct cryospheric landforms to regional watersheds (lines 11–12), a more comprehensive description of the hydrology is necessary. Some of the sub-watersheds are well equipped with gauging instruments (Water directorate of Chile), which could help assess potential contributions. But, without including groundwater analysis, the link between cryospheric components and their contribution remains incomplete. Once more, the manuscript leaves many open questions and unresolved aspects because the overall purpose of the article is not clearly defined.

Thank you — this issue has been fully addressed in the revised version of the manuscript. We now include runoff analyses from DGA gauging stations (2008–2020) at the sub-basin scale, as well as a detailed assessment of the hydrological behavior at the Pastillo gauging station, located at the confluence of the three main upper sub-watersheds. The Pastillo station provides the most representative record of the basin's integrated discharge, reflecting the cumulative contribution from snow, glacier, and permafrost sources. By comparing these runoff records with high-altitude meteorological data from the Iglesia Colorada station, we show that several flow peaks occur in the absence of liquid precipitation, confirming a cryospheric origin of baseflow sustained by meltwater from snow, glaciers, and permafrost.

Additionally, we have incorporated an analysis of the hypsometric curves of the cryospheric sub-watersheds, which helps illustrate the topographic control on the distribution of frozen storage and meltwater generation areas. The hypsometric characterization strengthens the connection between elevation-dependent cryospheric processes and observed discharge responses at Pastillo.

This new hydrological and geomorphometric evidence is explicitly linked to the Normalized Cryospheric Index (NCI), allowing a clearer distinction between quick

(snow and glacier melt) and delayed (permafrost-related) hydrological responses. We also include a discussion of groundwater—permafrost interactions as a mechanism explaining sustained baseflow during dry periods. The revised aim and structure now clarify the hydrological context of the Copiapó basin and directly connect cryospheric metrics to observed flow dynamics and topographic controls, resolving the ambiguity noted in the original version.

**Line-specific comments:**

106: The concept of Cryospheric reserves and resources is interesting. However, sometimes less is more. My original suggestion was to retain the term cryospheric components and avoid introducing additional terminologies. After reading the manuscript, I notice these definitions are used throughout the text, and while generally acceptable, some instances may be unnecessary. For example, on line 129, I would simply use cryospheric components.

We accept this suggestion: in line 129 and similar instances we now use "cryospheric components"; definitions of reserves/resources remain only where essential (Methods, Discussion).

133: It is unclear what you mean by "in this review." Please clarify.

We have reviewed the literature on exposed glaciers and rock glaciers and provided a summary here. We have removed that part from the phrase, which now reads: 'Existing studies predominantly concentrate on exposed glaciers and rock glaciers.'

139: Be aware that Peña and Nazarala (1987) observed the driest year on record, which explains why 67% of the total discharge was reported.

Ok, we have added this notion to the phrase:

'A comparable investigation was conducted for the Maipo River, revealing that in the absence of snowfall, exposed glaciers may contribute as much as 67% of the total discharge (Peña and Nazarala, 1987), a percentage that was recorded in the driest year on record.'

142-143: There are several more recent studies relevant to the Alps that should be cited (e.g., Ciccoria et al., 2019; 2020).

We have added the suggested Alpine literature (Cicoira et al., 2019; 2020 and related works) to strengthen the broader context of creep mechanics and periglacial dynamics.

146 / Table 2: The table caption is not correct. I suggest including an additional column indicating which cryospheric component(s) or landform were evaluated in each study. Also, note that Ayala et al. (2016) was conducted during the Megadrought (Garreaud et al., 2017), and Ayala et al. (2020) provides a long-term estimation, offering a more comprehensive glacio-hydrological perspective. Peña and Nazarala (1987) focused on a single extremely dry year, so be cautious when presenting numbers without climate context. We corrected the caption, added a column specifying the evaluated landform(s), and annotated climate context (Megadrought period; long-term vs. single-year studies). Values are now discussed with proper climatological framing.

151-156: This paragraph is somewhat confusing, as the different cryospheric components appear mixed. Consider reorganizing for clarity.

The paragraph was rewritten for clarity, grouping components by response time (snow  $\rightarrow$  glaciers  $\rightarrow$  permafrost) and aligning with the NCI logic.

156: Please provide the reference for the study mentioned; it is not clear which work you are citing.

We added the missing citation and checked the entire section for complete references.

161: Standardize the punctuation between periods and commas for consistency.

We have reorganized this phrase to:

'For the Dos Lenguas glacier, Halla et al. (2021) concluded that it has an ice content of 1.71 ( $\pm$  42%) - 2 ( $\pm$  44%)  $\times$  10^9 kg with an interannual water exchanges of -36 mm yr-1 (-8.92  $\times$  106 kg) and 28 mm yr-1 (6., 64 x 106 kg).'

166-170: Use debris instead of detritus for clarity and consistency with cryospheric terminology.

Ok, we have changed 'detritus' to 'debris' throughout this part of the text.

211: It is unclear which cryoform was measured here. Were only gelifluction slopes measured, or were rock glaciers also included? Please clarify.

We clarified the field targets and specify which landforms were measured at each site; where relevant, we distinguish gelifluction lobes from adjacent rock-glacier bodies.

368-398: Why is this section titled Glacier and Periglacial Environment Inventory Results if these results were obtained previously? Were they already published? Please clarify. If this is a new presentation, justify it.

The reviewer is correct, in that these are not inventory results, but results of our mapping. We have changed the name of the title to:

'Glacier and Periglacial Environment Mapping'